# Is One GPU Enough? Pushing Image Generation at Higher-Resolutions with Foundation Models.

**Athanasios Tragakis***
University of Glasgow
Glasgow, United Kingdom
a.tragakis.1@research.gla.ac.uk

**Marco Aversa***
Dotphoton
Zug, Switzerland
marco.aversa@dotphoton.com

**Chaitanya Kaul**
University of Glasgow
Glasgow, United Kingdom
chaitanya.kaul@glasgow.ac.uk

**Roderick Murray-Smith**
University of Glasgow
Glasgow, United Kingdom
Roderick.Murray-Smith@glasgow.ac.uk

**Daniele Faccio**
University of Glasgow
Glasgow, United Kingdom
Daniele.Faccio@glasgow.ac.uk

## Abstract

In this work, we introduce Pixelsmith, a zero-shot text-to-image generative framework to sample images at higher-resolutions with a single GPU. We are the first to show that it is possible to scale the output of a pre-trained diffusion model by a factor of 1000, opening the road for gigapixel image generation at no additional cost. Our cascading method uses the image generated at the lowest resolution as a baseline to sample at higher-resolutions. For the guidance, we introduce the Slider, a tunable mechanism that fuses the overall structure contained in the first-generated image with enhanced fine details. At each inference step, we denoise patches rather than the entire latent space, minimizing memory demands such that a single GPU can handle the process, regardless of the image's resolution. Our experimental results show that Pixelsmith not only achieves higher quality and diversity compared to existing techniques, but also reduces sampling time and artifacts.[1]

## 1 Introduction

Recent advances in diffusion models (DMs) have revolutionized the field of high-fidelity image generation. Foundational works like Ho et al. [2020], Sohl-Dickstein et al. [2015], Song and Ermon [2019] established the groundwork, leading to significant breakthroughs demonstrated by Dhariwal and Nichol [2021]. These models have evolved rapidly, with innovations such as new sampling techniques Lu et al. [2022], Song et al. [2020] and capabilities for inpainting Lugmayr et al. [2022] and image editing Brooks et al. [2023], Mokady et al. [2023], Nichol et al. [2021]. However, even though DMs have shown remarkable results on high-fidelity image generation, scaling them to high-resolutions is still an open challenge. The introduction of the latent diffusion model (LDM) Rombach et al. [2022b] has made high-resolution image synthesis more accessible. However, most pre-trained DMs based on the LDM are limited to generating images with a maximum resolution of

---

*Equal Contribution.

[1]The code for our work is available at https://thanos-db.github.io/Pixelsmith/.

38th Conference on Neural Information Processing Systems (NeurIPS 2024).

$1024^2$ pixels due to constraints in computational resources and memory efficiency. Following this approach, aiming to ultra-high-resolution generation would incur additional training and data costs, and the model would be unable to run on a single GPU.

Recently, several works have focused on scaling pre-trained models to higher-resolutions, highlighting new possibilities and challenges in the field Aversa et al. [2024], Du et al. [2023], Gu et al. [2023], He et al. [2023]. This allows already available models to be used with no extra costs and no additional carbon footprint. However, most methods addressing this problem either require expensive GPUs, as memory demands increase with resolution or prolonged generation times. Moreover, scaling the native resolution of a generative foundation model to higher-resolutions introduces artifacts due to the direct mapping into a prompt-image embedding space. For a specific prompt, we would expect to sample an image matching the description and of the same size as the training data images. Consequently, even scaling up by a factor of 2 would lead to the duplication of the image produced by the prompt across the higher-resolution image. However, since the images in the training set contain information at different resolutions, we can leverage this prior knowledge implicitly learned by the diffusion model to guide the generation at higher-resolutions and enhance fine details. Addressing these challenges is crucial for applications that demand ultra-high-resolution images, such as gigapixel photography, medical imaging, satellite imagery and high-definition digital art.

To overcome the challenges in ultra-high-resolution image generation, we introduce Pixelsmith, an adaptable framework that utilizes pre-trained generative models for scalable gigapixel synthesis.

Our contributions are outlined as follows: 1. We introduce Pixelsmith, the first framework capable of generating gigapixel-resolution images using a single GPU. 2. We develop the Slider, a dynamic tool that allows users to adjust the balance between overall image structure and fine-detail enhancements in the generation process. 3. We enhance and adapt the random patch denoising strategy to text-to-image pre-trained diffusion models, leading to minimized memory usage. 4. We provide a masking method that, combined with the Slider, reduces the number of artifacts at higher-resolutions.

## 2    Related Work

Pre-trained DMs are trending toward increasing native resolutions. Notable examples include Stable Diffusion (SD) Rombach et al. [2022b], which started at $512^2$, then $768^2$ in SD 2, and $1024^2$ in SDXL Podell et al. [2023], SD Cascade Pernias et al. [2024], and SD 3 Esser et al. [2024]. Similarly, DALL-E Ramesh et al. [2021] continues to increase resolution with each version OpenAI [Nov 6 2023]. This trend shows a growing demand for higher-resolution generation.

Currently, high-resolution image generation often involves a super-resolution model applied after the initial text-to-image generation Saharia et al. [2022]. This additional model increases costs due to training and domain-specific fine-tuning requirements.

### 2.1    Trained Models

Trained models are those that are specifically designed for multi-resolution generation. Recent models like Matryoshka Gu et al. [2023] can generate various resolutions up to $1024^2$ through a progressive training schedule. However, scaling beyond this is limited. CogView3 Zheng et al. [2024b], based on Relay Teng et al. [2023], upsamples from a base resolution of $512^2$, though higher-resolutions like $4096^2$ remain a future goal. Fine-tuning methods such as DiffFit Xie et al. [2023] are costly, requiring 51 V100 GPU days. ASD Zheng et al. [2024a] introduces a memory-efficient sampling method capable of *theoretically* generating images up to $18432^2$ resolution. Patch-DM Ding et al. [2023], by training on $64^2$ patches, can generate resolutions like 1024×512. LEGO Zheng et al. [2023] and Inf-DiT Yang et al. [2024] also rely on training.

### 2.2    Adapted Models

Adapted models, on the other hand, modify pre-trained models to generate higher-resolutions without additional training. MultiDiffusion Bar-Tal et al. [2023] offers controllable generation through multiple processes but is slow and prone to structural errors Du et al. [2023], He et al. [2023], Zheng et al. [2024a]. ScaleCrafter He et al. [2023] improves resolution by altering the convolution kernel dilation but faces memory limitations. DemoFusion Du et al. [2023] adapts MultiDiffusion

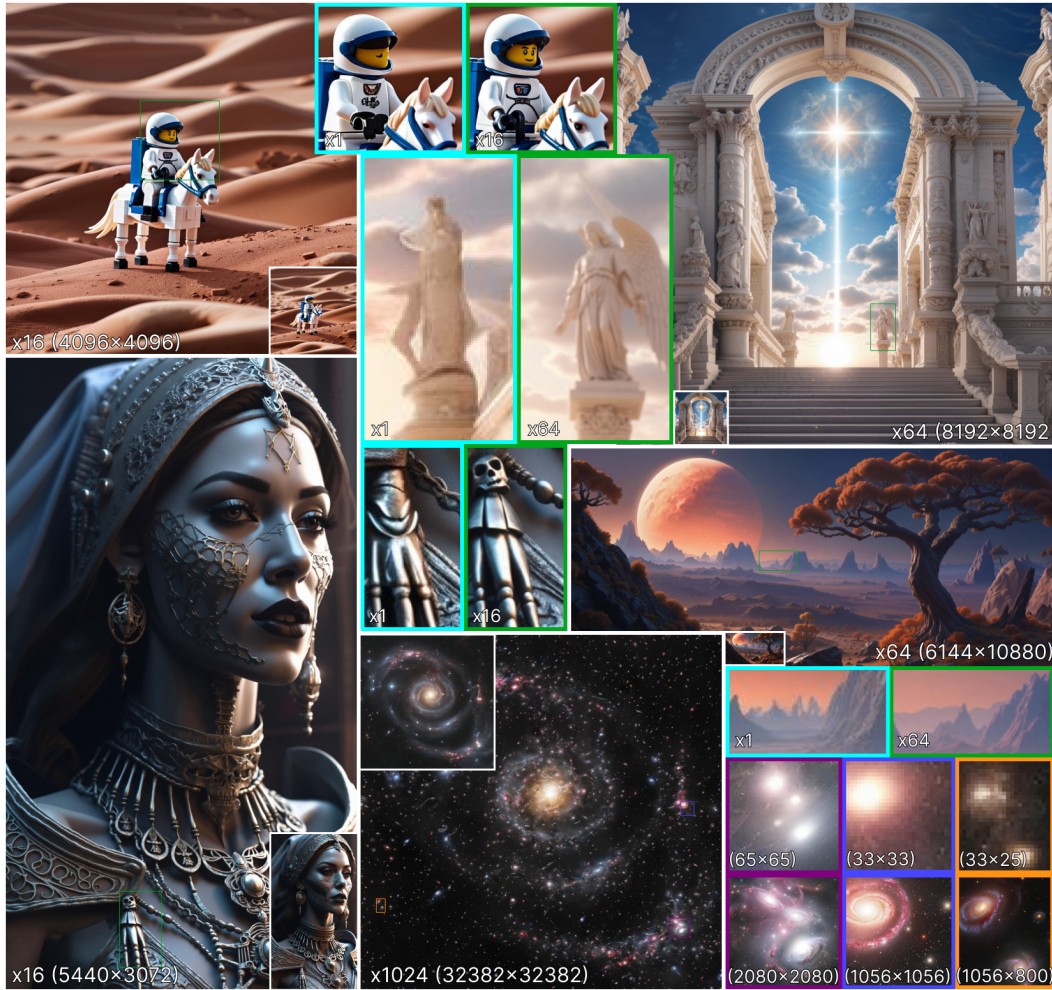

Figure 1: Examples of generated images using Pixelsmith. The proposed framework generates images on higher-resolutions than the pre-trained model without any fine-tuning. Images at different resolutions are shown with cut-out areas for both Pixelsmith and the base model. The higher-resolution images are in scale with the images generated by the base model. Only the lower resolution version of the gigapixel image has been resized for a better visualisation. Some cut-outs of the gigapixel generation have resolution close to the base model which is $1024^2$ and it can be seen that the images are comparable in aesthetics showing that our framework is capable of true gigapixel generations (**zoom in** to see in better detail).

for constant memory use, though it is time-consuming. ElasticDiffusion Haji-Ali et al. [2023] and SyncDiffusion Lee et al. [2023] aim for specific high-resolution goals, but object repetition remains a problem at large scales Jin et al. [2024]. Recently, HiDiffusion Zhang et al. [2023] introduced modifications to the UNet architecture to prevent object duplication; however, it is limited to a maximum resolution of $4096^2$ due to high GPU memory requirements. AccDiffusion Lin et al. [2024] employs patch-content-aware prompts and adjusts the attention masks within the UNet to suppress artifacts, but these changes result in blurry images. Similarly, Fouriscale Huang et al. [2024] is another model that alters the UNet but exhibits issues at higher-resolutions, particularly at $4096^2$. Current approaches excel in some areas but fall short in others. To enable high-resolution generation on a single consumer GPU, models must be adapted to avoid retraining costs. Efficient memory usage is crucial to prevent the need for expensive, high-memory GPUs, and the process must be fast and artifact-free. We propose a flexible framework that addresses these challenges based on text conditions.

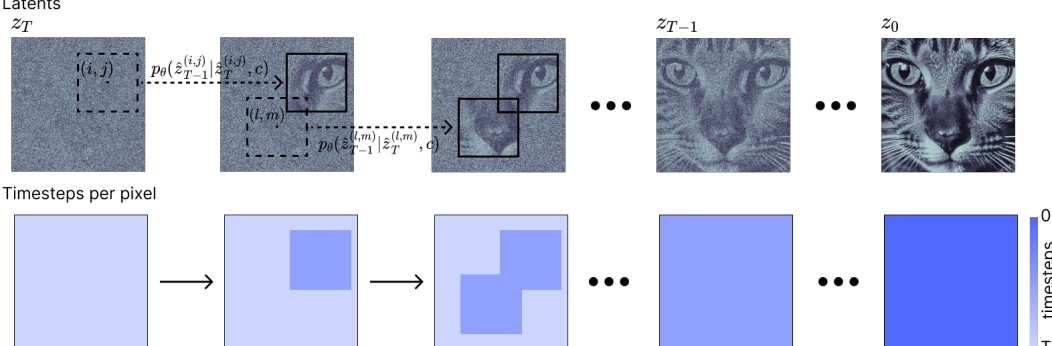

Figure 2: Overview of the patch denoising process proposed by DiffInfinite: The top row represents the latent space, while the bottom row tracks the timesteps for each pixel. Each pixel should be denoised only once per timestep, so when overlapping occurs, already denoised pixels revert to their previous values from the prior timestep. After denoising, these reverted pixels are restored to their original denoised state from the current timestep.

# 3 Foundations

## 3.1 Diffusion models

DMs Ho et al. [2020], Sohl-Dickstein et al. [2015] are probabilistic generative models that first add noise to a distribution during diffusion and then learn to remove this noise during denoising. This way, during training, a Gaussian probability distribution is learnt and during inference sampling from the Gaussian leads to the data probability distribution. Executing this process in the latent space Rombach et al. [2022b] is more resource efficient allowing for faster training and inference times. In a Latent Diffusion Model, let $z_0 \equiv \mathcal{E}_\theta(x)$ represent the clean training data $x \sim q(x)$ mapped to the latent space by the VQ-VAE encoder $\mathcal{E}_\theta$. Beginning from $z_0$, at each timestep $t \in \{1, \ldots, T\}$, noisy latents $z_1, \ldots, z_T \sim q(z_t \mid z_{t-1}) := \mathcal{N}(z_t; \sqrt{1 - \beta_t} z_{t-1}, \beta_t \mathbf{I})$ are sampled according to a noise schedule function based on the timestep $\beta_t \in [0, 1]$. During training, the model learns to denoise these latent variables using a conditional prompt $c$. At inference time, starting from a random latent $z_T \sim \mathcal{N}(0, \mathbf{I})$, the model generates by iteratively sampling $z_0 \sim p_\theta(z_0 \mid z_T, c) = \prod_{i=1}^{T} p_\theta(z_{t-1} \mid z_t, c)$.

## 3.2 Patch sampling

The default LDM denoising process samples the entire latent space at each timestep, which becomes resource-intensive as the resolution increases. Methods like Li et al. [2024] distribute this load across multiple GPUs, but this requires expensive hardware. To address this, we adapted and refined the DiffInfinite sampling method Aversa et al. [2024] for text-to-image DMs, enabling ultra-high-resolution generation efficiently on a single GPU.

At each timestep, random patches are selected for denoising, and this process is repeated until the entire latent space is denoised. The randomness is controlled for efficiency. We track which pixels have been denoised at each timestep, and select areas where pixels have not been denoised yet. This sampling process avoids excessive inference times, especially at very high-resolutions like $32768^2$. In our experiments, we sample images at higher-resolution using fixed $128^2$ patches in the latent space. To handle overlapping patches, pixel values are temporarily reverted when already denoised pixels are encountered, ensuring correct processing. After the current patch is denoised, the pixels that were reverted are restored to their denoised values. Only the first denoising of each pixel is retained for each timestep, preventing redundant denoising (see Fig. 2).

Despite its advantages, the DiffInfinite sampling method still relies on segmentation masks to condition each patch, providing rich spatial information. However, in text-to-image DMs, where a global text prompt conditions the entire latent space, this method is insufficient on its own. Applying the same text prompt to each patch results in repetitive content and poor-quality generations. To overcome this, we introduce a guiding mechanism that incorporates structural information between patches alongside the text prompt, improving the quality and diversity of the generated images.

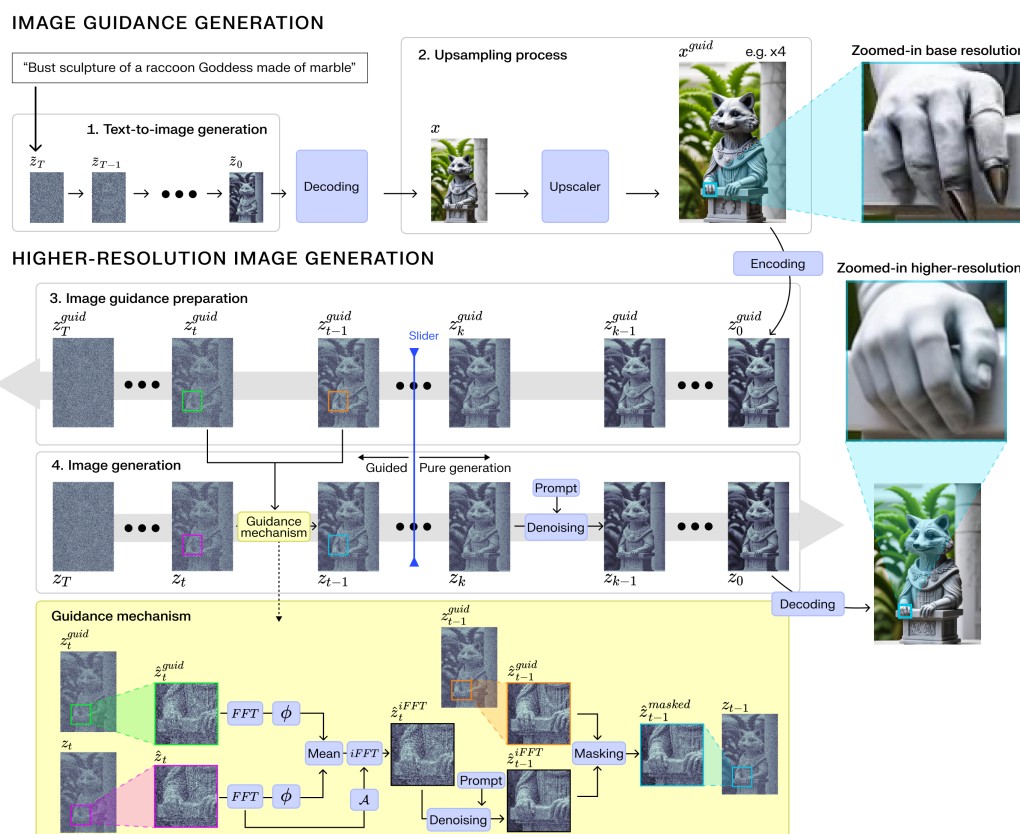

Figure 3: Proposed framework overview. 1. *Text-to-image Generation*: A pre-trained text-to-image diffusion model generates an initial image based on the input text prompt. 2. *Upsampling process*: The generated image is upscaled (in this use case by a factor x4) and encoded into the latent space to guide the creation of a higher-resolution image. 3. *Image guidance preparation*: The encoded image is degraded through the diffusive forward model, creating the guidance latents. 4. *Image generation*: the Slider (indicated by a blue line) adjusts the extent of guidance. *Left of Slider* (Guided Generation): guidance latents control the image generation. The framework fuses guidance latents (green patches) with high-resolution latents (purple patches) using the Fast Fourier Transformation (FFT). The phases are averaged and combined with the amplitude, then transformed back via the inverse FFT (iFFT). A chess-like mask integrates information from the successive guidance step (orange), resulting in fully processed patches (cyan). *Right of Slider* (Pure Generation): the generation relies only on the prompt. *Higher-Resolution Comparison*: while the base model upscales the bust with disfigured hands, the proposed method enhances details, corrects distortions, and prevents new artifacts.

## 4 Method

### 4.1 Problem statement

Currently, pre-trained text-to-image LDMs generate a latent variable $z_0 \sim p_\theta(z_0|z_T, c)$ with a fixed size using a diffusion process, which is then decoded into pixel space using the decoder $\mathcal{D}_\theta$ of a VAE. In this section, we show how to leverage foundational generative models to sample arbitrarily large images based on a given prompt. Without requiring additional training or fine-tuning, the LDM, trained on images with size $H \times W \times 3$, is adapted to generate images with size $mH \times nW \times 3$, where $m, n \geq 1$ are the scaling factors for achieving higher-resolution.

We present Pixelsmith, a framework with a flexible number of cascaded steps (Section 4.3) designed to generate images at ultra-high-resolution on a single GPU. The implementation of the Slider (Section 4.4) provides control over the generation process using patches (Section 4.2), making it resource-efficient regardless of the resolution.

## 4.2 Framework overview

In this section, we describe the workflow of Pixelsmith, detailing how the framework adapts pre-trained text-to-image LDMs to generate images with higher-resolutions on a single GPU (see Fig. 3). In order to generate ultra-high-resolution images without artifacts, we introduce these key components: the Slider (see Sec. 4.4), patch averaging (see Sec. 4.5) and masking (see 4.6).

**Text-to-image generation** First, given a conditional prompt $c$, we use SDXL to sample the latent variable $\tilde{z}_0$ and decode it into pixel space as $\tilde{x} = \mathcal{D}_\theta(\tilde{z}_0)$, generating a $1024^2$ resolution image.

**Upsampling process** After the image generation, we apply an upsampling algorithm to increase the image resolution (see Sec. 4.3 for details). In our case, we used Lanczos interpolation Lanczos [1950] to scale up to the desired resolution. However, upsampling leads to a blurred output and lack of additional content. The upsampled image, $x^{guid}$, will serve as guidance for our generative process.

**Image guidance preparation** Once the guidance image is encoded in the VAE's latent space $z_0^{guid} = \mathcal{E}_\theta(x^{guid})$, we can easily sample each latent variable of the diffusion process through the forward diffusion process $z_t^{guid} \sim q(z_t^{guid}|z_0^{guid})$.

**Image generation** The generative process starts from $z_T \sim \mathcal{N}(0, I)$, which has the same dimensions as $z_T^{guid}$. At each step, a random patch is cropped as described in Section 3.2:

$$\hat{z}_t^{(i,j)} = \mathcal{C}^{i,j}(z_t), \quad \hat{z}_t^{guid,(i,j)} = \mathcal{C}^{i,j}\left(z_t^{guid}\right) \tag{1}$$

where $\mathcal{C}^{i,j} : \mathbb{R}^{4,h,w} \to \mathbb{R}^{4,p_h,p_w}$ is a cropping function that extracts patches of size $(p_h, p_w)$ from the latent variables $z_t$ and $z_t^{guid}$ of size $(h, w)$ at the coordinates $(i, j)$. For simplicity, we will refer to the cropped latent patch $\hat{z}_t^{(i,j)}$ as $\hat{z}_t$ and $\hat{z}_t^{guid,(i,j)}$ as $\hat{z}_t^{guid}$ throughout this discussion.

The Slider's position (see Sec. 4.4 for details), indicated by a blue line in Fig. 3, determines whether the guidance mechanism (see Sec. 4.4.1 for details) or unguided patch denoising will be applied. In the unguided mode, each patch is based solely on the previous one and the text condition, similar to a conventional patch denoising process. The Slider allows control over whether a generated image will be slightly or significantly altered compared to the previous resolution.

After the denoising process has ended, the latents $z_0$ are decoded and the higher-resolution image is generated. Using a cascade upsampling approach, the generated image can be upsampled again, repeating the process to achieve an even higher-resolution image.

## 4.3 Higher generation in one step

Most of existing higher-resolution generative methods rely on cascade sampling approaches Denton et al. [2015], Ho et al. [2022], Menick and Kalchbrenner [2018]. Our approach, however, differs from previous works like Du et al. [2023], Guo et al. [2024] in two significant ways.

First, we perform upsampling directly in the pixel space rather than in the latent space. Manipulating the latent space distribution with transformations like upsampling can introduce distortions that the scheduler has not accounted for Chang et al. [2024], leading to degraded image quality Hwang et al. [2024]. Second, our flexible method enables higher-resolution image generation in two generative steps: first by producing the base image, then by enhancing it to the desired resolution. For instance, Du et al. [2023] requires passing through intermediate resolutions ($1024^2$, $2048^2$, $3072^2$) to achieve a $4096^2$ resolution, with even higher-resolutions necessitating additional steps. This results in prolonged waiting times and multi-scale duplications, as each step may produce duplicates at different resolutions (Appendix D).

In contrast, our framework can generate any resolution directly from the base one, making it feasible to scale up to a $32768^2$ image directly from the base resolution of $1024^2$. However, intermediate steps can sometimes yield better small-sized details (Appendix E). Our flexible approach supports both two-step generation and cascade generation, allowing users to select their preferred sampling method depending on the use case.

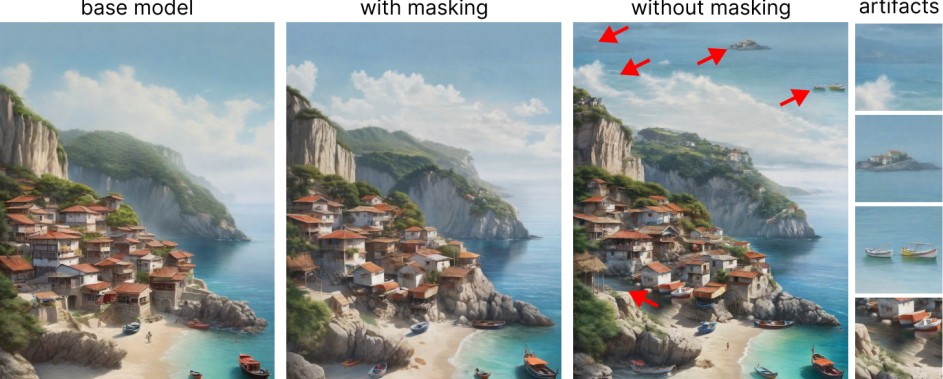

base model    with masking    without masking    artifacts

Figure 4: Masking effects on higher-resolution generation ($\times 16$ the original resolution). (left) Image generated using SDXL. (center) Image generated with Pixelsmith with masking. (right) Image generated with Pixelsmith without masking. We highlighted the artifacts introduced by generating at higher scales. These artifacts demonstrate the challenges of maintaining coherence and accuracy when scaling up the resolution without additional guidance.

### 4.4 Slider

Currently, higher-resolution generative models often encounter issues such as duplications and atypical anatomical features or structures He et al. [2023]. To mitigate these artifacts, we introduce the *Slider* (indicated by a blue line in Figure 3). The Slider is a parameter that determines at which denoising step there is the transition between the guidance mechanism introduced in Sec. 4.4.1 and traditional unguided generation. The Slider can either constrain the denoising process to eliminate undesirable outputs or allow for less constrained generation to add details.

When set to zero, the generation is entirely unconstrained. This results in an image that might resemble the $1024^2$ base image but will likely contain multiple duplicates. At $t = T$, the generation is highly guided, leading to an image that is almost identical to the previous resolution. The optimal value for the Slider varies depending on the base image. Adjusting the Slider manually is necessary to achieve the best output. This manual adjustment allows for fine-tuning the balance between constrained and unconstrained generation, ensuring that the resulting image maintains the desired quality and level of detail. The ideal Slider setting is influenced by factors such as image complexity, the presence of fine details, and the desired resolution. Experimentation and iterative adjustments are often required to find the most effective value for each specific image and resolution.

An example demonstrating the impact of the Slider's position can be found in Appendix G.

#### 4.4.1 Guidance mechanism

The guidance mechanism combines the random patches $\hat{z}_t^{\text{guid}}$, $\hat{z}_t$, and $\hat{z}_{t-1}^{\text{guid}}$ to generate the updated patch $\hat{z}_{t-1}$. First, the patches $\hat{z}_t^{\text{guid}}$ and $\hat{z}_t$ are transformed into the Fourier space using the Fast Fourier Transform ($\mathcal{FFT}$), where their phase components $\phi_t$ and $\phi_t^{\text{guid}}$ are averaged:

$$\begin{cases} \mathcal{FFT}\left(\hat{z}_t\right) = \mathcal{A}_t e^{i\phi_t} \\ \mathcal{FFT}\left(\hat{z}_t^{\text{guid}}\right) = \mathcal{A}_t^{\text{guid}} e^{i\phi_t^{\text{guid}}} \end{cases} \Rightarrow \quad \overline{\phi}_t = \arg\left(e^{i\phi_t} + e^{i\phi_t^{\text{guid}}}\right) \tag{2}$$

Here, $\mathcal{A}_t$ and $\mathcal{A}_t^{\text{guid}}$ represent the amplitudes, while $\phi_t$ and $\phi_t^{\text{guid}}$ are the phases.[2] The averaged phase $\overline{\phi}_t$ is then combined with the amplitude $\mathcal{A}_t$ from $\hat{z}_t$ to form a new Fourier representation $\hat{z}_t^{\text{FFT}} = \mathcal{A}_t e^{i\overline{\phi}_t}$. While the amplitude $\mathcal{A}_t$ preserves the intensity and contrast information, the phase averaging process ensures that the resulting phase contains structural characteristics from both

---

[2]Directly averaging phases, as in $\frac{\phi_1 + \phi_2}{2}$, can lead to issues due to the periodic nature of sinusoidal functions. For example, averaging $\phi = 0$ and $\phi = 2\pi - \epsilon$ should result in a phase near zero, not $\pi$. To correctly average phases, we compute the $arg(\cdot) = \arctan(y/x)$ of the sum of the complex numbers' phase components.

the generated and guiding images. This modified Fourier representation is transformed back to the spatial domain using the inverse Fast Fourier Transform $\hat{z}_t^{iFFT} = i\mathcal{FFT}\left(\hat{z}_t^{FFT}\right)$. The output $\hat{z}_t^{iFFT}$ is then used as the condition for the reverse diffusion process to generate the next patch $\hat{z}_{t-1}^{iFFT} \sim p_\theta\left(\hat{z}_{t-1}^{iFFT} \mid \hat{z}_t^{iFFT}, c\right)$. This solution suppresses local artifacts and helps maintain low-frequency structural consistency between the reverse diffusion steps, reducing the likelihood of significantly altering the global properties of the generated image. However, the guidance mechanism alone still generates some long-range discrepancies across the image due to prompt sharing between the patches. To mitigate these artifacts and prevent prompt duplications across the latent space, we introduce a masking method(see Section 4.6).

## 4.5 Patch averaging

Overlapping patches can sometimes cause visible differences at their borders. To address this, we introduce a transition zone where the values of overlapping patches at timestep $t$ are averaged, producing a smooth and seamless denoised output. In Appendix C, we illustrate the artifacts caused by patch overlap and show how averaging effectively removes them.

## 4.6 Masking

One of the main reasons patch-based image generation in diffusion models Bar-Tal et al. [2023] suffers from artifacts at higher-resolutions is the use of the overall image text prompt for each patch during the denoising process. This leads to duplicate structures forming in the latent space. We partially addressed the spatial coherence with the guidance mechanism in Section 4.4.1. To further improve results, we combine the sampled $\hat{z}_t^{iFFT}$ with the image guidance $\hat{z}_t^{guid}$ using a chess-like mask $\Lambda$ (Equation 3).

$$\hat{z}_t^{masked} = \Lambda\hat{z}_t^{iFFT} + (1-\Lambda)\hat{z}_t^{guid} \quad \text{where } \Lambda = \lambda_{i,j} = \begin{cases} 0 & \text{if } i+j \text{ is even} \\ 1 & \text{if } i+j \text{ is odd} \end{cases} \quad (3)$$

In Fig. 4, we show the comparison between a higher-resolution image generated with and without masking with the image guidance. The image on the right, which is not constrained by pixels from $\hat{z}_t^{guid}$, exhibits more freedom in generation and, as a result, introduces notable artifacts. For example, clouds transform into waves, and the blue sky morphs into the sea. Additionally, a small, seemingly artificial lake or river appears in the middle of the village, situated unnaturally close to the beach.

# 5 Experiments

Pixelsmith is tested on a single RTX 3090 GPU, with all tested resolutions requiring 8.4 GB of memory. Performance is evaluated on the LAION-5B dataset Schuhmann et al. [2022] by randomly sampling 1,000 image and text prompt pairs. The metrics used for evaluation are Fréchet Inception Distance (FID) Heusel et al. [2017], Kernel Inception Distance (KID) Bińkowski et al. [2018], Inception Score (IS) Salimans et al. [2016], and CLIP Score Radford et al. [2021], with the FID metric computed using the clean-FID approach Parmar et al. [2022] (for further comparisons, see Appendix B). Additionally, we compare our results with a super-resolution model in Appendix F.

## 5.1 Ablation study

We conducted a quantitative examination of our framework at a resolution of $2048 \times 2048$ pixels, focusing on key factors that influence its performance: the Slider position, the role of amplitude and phase in the latent space, the importance of masking during guidance, and the impact of averaging overlapping patches. Our results are shown in Table 1.

**Slider position** Our findings highlight the critical role of the Slider position in the quality of the generated images. Setting the Slider to position 30 (*proposed* method) provides an optimal trade-off between guidance and generation, outperforming other positions like 1 (*SP1*, no guidance), 24 (*SP24*, mid-point), and 49 (*SP49*, full guidance). A Slider position of 1 introduces numerous artifacts due to insufficient guidance, while a position of 49 lacks fine detail because it relies too heavily on lower-resolution. Position 24 has been chosen as mid-point in the diffusion process, it is closer to the optimal value but does not yield the best results across our dataset.

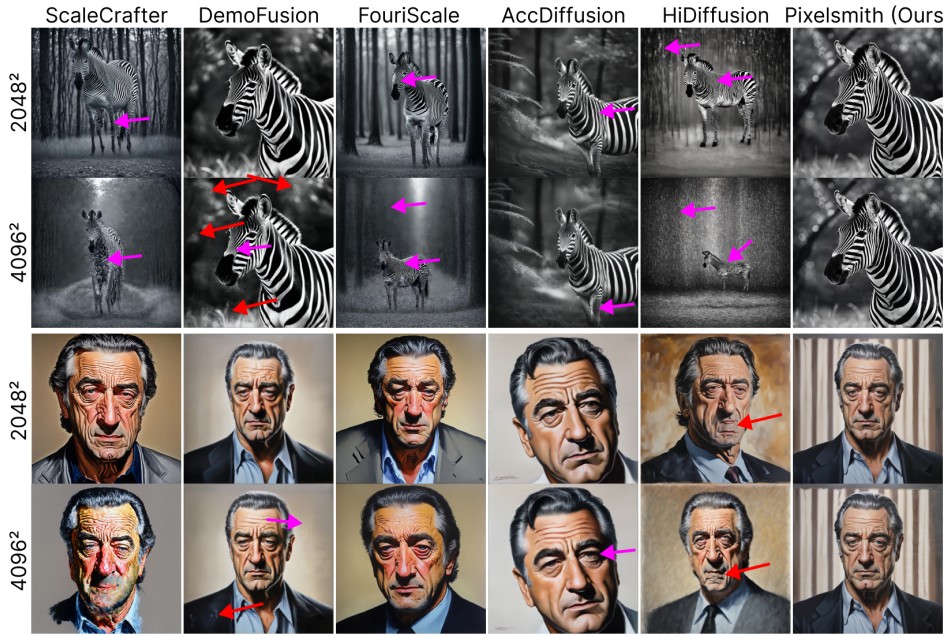

Figure 5: Qualitative comparisons: This figure highlights how other models suffer from duplications (red arrows) and introduce artifacts in areas with complex, high-frequency patterns (purple arrows). In contrast, Pixelsmith effectively eliminates these issues. (**zoom in** to see in better detail).

**Amplitude and phase**    In exploring the role of amplitude and phase within the latent space, we experimented with different configurations. The *proposed* method averages the phase of the guidance latents and the current latents while using the amplitude of the current latents, as detailed in the Methods section (see Figure 3). We compared it with other setups: one that averages the amplitude of both latent spaces while using the phase of the current latents (denoted as $\mathcal{A}$ in Table 1), and another that averages both amplitude and phase from both latent spaces (denoted as $\mathcal{A}\&\phi$). The proposed method demonstrated superior performance in maintaining image quality and preserving fine details. Averaging the two phases provides guidance through structural information, while averaging the amplitudes only conveys the intensity of the frequencies. Averaging both offers information on both structure and intensity which overly influences the final image, leading to a lack of fine details expected in a higher-resolution output.

**Masking**    Masking emerges as a significant factor in enhancing image quality. The quantitative metrics indicate that the chess-like mask leads to slightly better scores compared to using no mask (as seen in the *No Mask* column of Table 1 compared to the *proposed* one). Visual inspections reveals that incorporating the mask is crucial for removing artifacts (see Figure 4).

**Overlapping patches**    Lastly, we assessed the impact of averaging overlapping patches. Implementing patch averaging improved the performance compared to not using it, as evidenced by the comparison between the *proposed* method and the *No Averaging* column in Table 1. Although, despite the small difference in the metrics, averaging overlapping patches effectively eliminates artifacts that occur at patch borders due to overlapping regions, resulting in smoother and more coherent images (see Appendix C). This discrepancy underscores the limitations of metrics like FID in detecting high-resolution artifacts, highlighting the necessity of qualitative assessments.

Table 1: A quantitative examination of our framework through ablations.

| Metric | SP1 | SP24 | SP49 | $\mathcal{A}$ | $\mathcal{A}\&\phi$ | No Mask | No Aver. | Proposed |
|---|---|---|---|---|---|---|---|---|
| FID↓ | 74.890 | 63.168 | 64.195 | 63.476 | 63.173 | 64.473 | 63.493 | **63.116** |
| KID↓ | 0.008 | **0.002** | 0.003 | **0.002** | **0.002** | 0.003 | 0.003 | **0.002** |
| IS↑ | 19.928 | 19.941 | 19.290 | 19.933 | 19.897 | 19.892 | 19.626 | **24.242** |
| CLIP↑ | 32.198 | 32.451 | 32.425 | 32.730 | 32.432 | 32.355 | 32.416 | **33.429** |

## 5.2   Comparison

We compare Pixelsmith to state-of-the-art frameworks in the higher-resolution image generation task, as well as to SDXL, which serves as the base model for our adaptation. To ensure a fair comparison, we include only models with native resolutions comparable to ours, excluding those trained at resolutions smaller than $1024^2$. While Pixelsmith outperforms other methods in most metrics (see Table 2), its real advantage lies in its ability to adapt to the unique characteristics of each individual image generation. This flexibility enables precise control to emphasize details and reduce artifacts. In Figure 5, we qualitatively show how Pixelsmith maintain the image structure by adding fine-details while other models introduce duplications or artifacts. There, we highlighted with red arrows the duplications and with purple arrows spurious artifacts. Additional comparisons can be found in Appendix B.

Table 2: Quantitative comparisons with existing works.

| Resolution | Model | FID↓ | KID↓ | IS↑ | CLIP↑ | Time (sec)↓ |
|---|---|---|---|---|---|---|
| $2048^2$ | SDXL Podell et al. [2023] | 111.452 | 0.020 | 12.046 | 29.804 | 71 |
| | ScaleCrafter He et al. [2023] | 77.543 | 0.006 | 17.112 | 30.904 | 80 |
| | DemoFusion Du et al. [2023] | 64.422 | **0.002** | 19.307 | 32.648 | 219 |
| | HiDiffusion Zhang et al. [2023] | 74.773 | 0.0051 | 17.572 | 31.250 | **50** |
| | FouriScale Huang et al. [2024] | 75.378 | 0.007 | 17.987 | 31.028 | 162 |
| | AccDiffusion Lin et al. [2024] | 65.728 | 0.004 | 19.776 | 31.789 | 231 |
| | Pixelsmith (**Ours**) | **63.116** | **0.002** | **24.242** | **33.429** | 130 |
| $4096^2$ | SDXL Podell et al. [2023] | 195.117 | 0.069 | 7.709 | 24.565 | 515 |
| | ScaleCrafter He et al. [2023] | 105.132 | 0.018 | 13.542 | 27.767 | 1257 |
| | DemoFusion Du et al. [2023] | 66.186 | 0.003 | 18.940 | 32.319 | 1632 |
| | HiDiffusion Zhang et al. [2023] | 97.614 | 0.015 | 13.681 | 27.708 | **255** |
| | FouriScale Huang et al. [2024] | 125.390 | 0.028 | 11,837 | 26.802 | * |
| | AccDiffusion Lin et al. [2024] | 67.084 | 0.003 | 19.323 | 32.010 | 1710 |
| | Pixelsmith (**Ours**) | **63.686** | **0.002** | **19.741** | **32.369** | 549 |

* Inference time not available for FouriScale at $4096^2$ resolution due to out of memory on an RTX 3090.

## 6   Discussion and Considerations

Pixelsmith can generate images at arbitrarily high-resolutions, but as the resolution increases, adding more generative details without introducing artifacts becomes difficult due to the smaller denoising patch size relative to the latent space. For example, a $32768^2$ image maps to a $4096^2$ latent space with a $128^2$ denoising patch. The higher the resolution to generate in a single step, the higher the Slider value is required to suppress artifacts. However, increasing the Slider value reduces the finer detail generation and return an image which closely resembles the previous resolution, limiting the true quality of the higher-resolution image. Generating an image through a cascade generation approach allows for using a lower Slider value at each stage, providing more details and fewer artifacts compared to single-step generation, as the lower Slider settings can be more effective in this incremental process. Additionally, appropriate metrics for evaluating high-resolution images are lacking, and further research is necessary to improve quantitative evaluation methods, as highlighted in previous studies Jin et al. [2024] as well.

## 7   Conclusions

In this work, we introduced Pixelsmith, a text-to-image generative framework that leverages a pre-trained foundation model to generate higher-resolution images. Our flexible method uses a base image generated at the lowest resolution, combined with the Slider mechanism, to ensure both structural coherence and fine detail enhancement in the resulting images. By employing patch-based denoising, we have significantly reduced memory demands, making the generation of ultra-high-resolution images feasible on consumer-grade GPUs. Experimental results demonstrate that our model outperforms current state-of-the-art methods in both image quality and generation efficiency.

# 8 Acknowledgements

D.F., A.T. acknowledge support from Royal Academy of Engineering through the Chairs in Emerging Technologies scheme. D.F., R.M-S., C.K. acknowledge funding from *QuantIC* Project funded by EPSRC Quantum Technology Programme (grant EP/MO1326X/1, EP/T00097X/1), and *Google*. R.M-S, C.K. acknowledge funding from EP/R018634/1, EP/T021020/1, and EP/Y029178/1. For the purpose of open access, the author(s) has applied a Creative Commons Attribution (CC BY) license to any Accepted Manuscript version arising.

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

# A Improving the base generation

We show the qualitative improvements Pixelsmith provides to the base model through an example here. We observe that Pixelsmith improves the generation at higher image generation resolutions. In Figure 6, we can see that relatively small and larger areas in the image are also improved after each iteration of the image generation. The final image resembles the actress more in terms of facial attributes and has more finer details in terms of multiple physical characteristics.

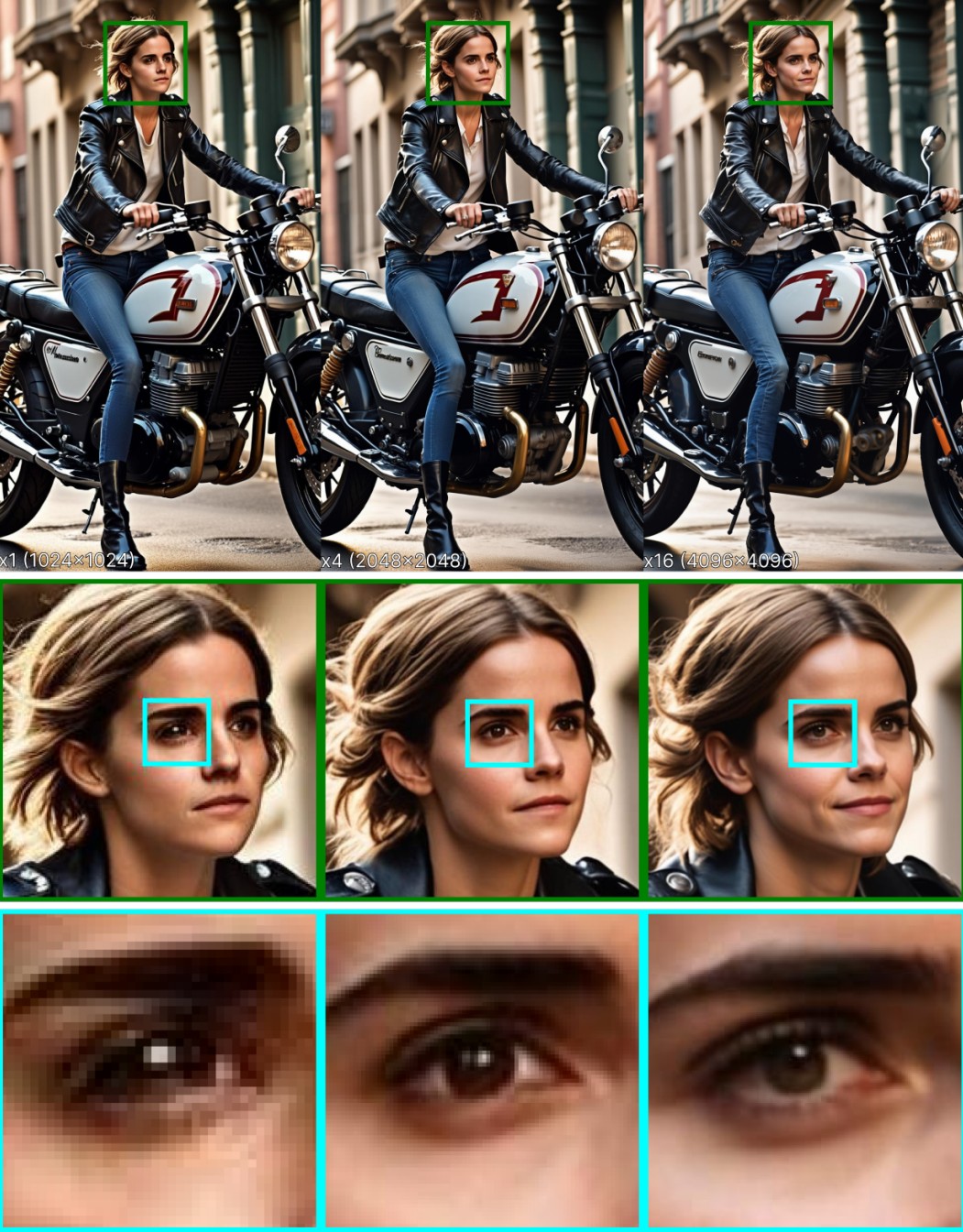

Figure 6: We show an example of a two-step image generation process here. The base model provides a $1024^2$ image which our method then uses to iteratively generate a $2048^2$ and a $4096^2$ image. It is evident from this example that the higher-resolutions overall have increasingly improved facial characteristics, demonstrating the effectiveness of our framework (**zoom in** to see in better detail).

# B    Additional comparisons

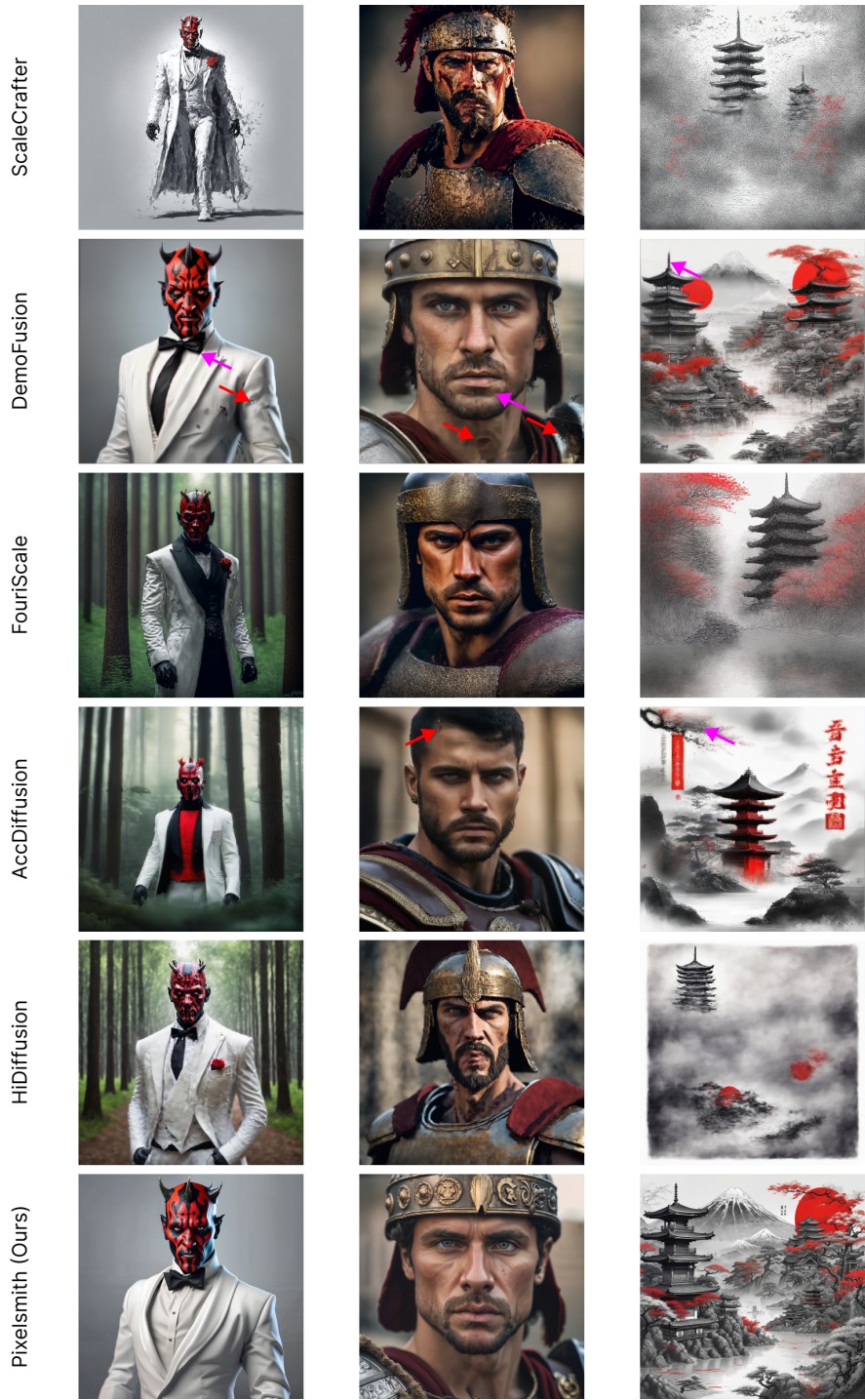

Figure 7: We show additional qualitative comparisons against our competitors here. All generated images have a resolution of $4096^2$. Other models suffer from artifacts (red arrows) along with highly prominent high frequency noise (purple arrows), *some* of which are indicated by arrows (**zoom in** to see in better detail).

## C  Averaging Overlapping Patches

Patch denoising may introduce artifacts as seen in Figure 8. This occurs because we sample overlapping patches from the latent space, which can result in parts of the same area being denoised differently across patches. We address this issue by averaging along the edges where the patches meet.

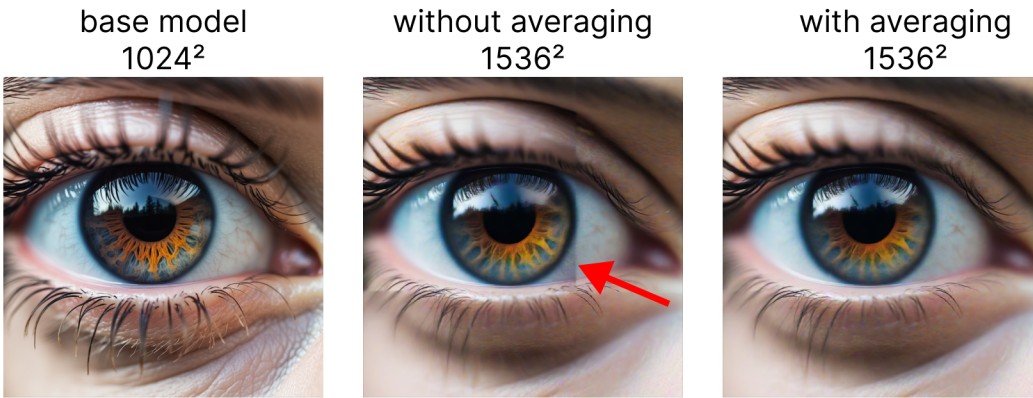

Figure 8: An illustration of a patch artifact. The left image with resolution $1024^2$ is generated by the base model. We generate the middle and right images at a resolution of $1536^2$. In the middle image an artifact is visible due to overlapping patch edges. By averaging the overlapping patches, such artifacts are removed, as seen in the image on the right (**zoom in** to see in better detail).

Figure 9 demonstrates our approach visually. Here, the image on the left represents the first noised latents. During the first denoising step (the next image), we sample a random patch (shown in blue) for the denoising process. After every denoising step, our algorithm checks for overlap between patches to perform averaging to remove any potential artifacts that may occur. The second denoising step (the third image), demonstrates a random patch sampled (in green) which overlaps with the patch in the previous denoising step. The third image shows an area of overlap where artifacts can be potentially generated. To alleviate this issue, we then take a tolerance of 10 pixels in the overlap but within the bounds of the first (blue) patch, and take an average across that region (as shown in yellow in the final image in the figure). From our experiments, we note that averaging across the edges of the patches, forces that region to create a smooth structure instead of horizontal and vertical lines (as shown in the second image in Figure 8).

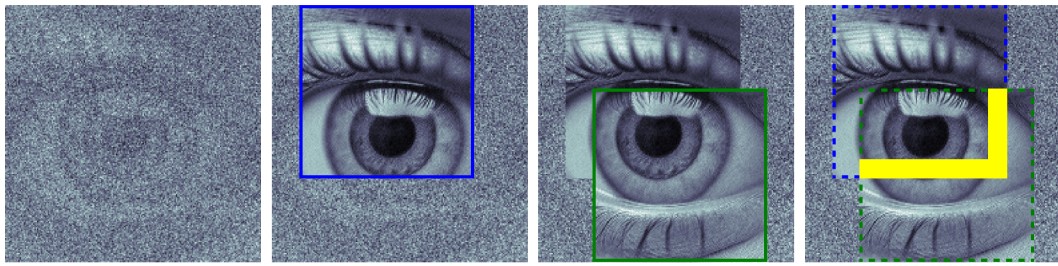

Figure 9: An example of overlapped patches. The left image shows the noised latents. The second image is the first denoising step, and the third is the second denoising step. The third image shows the area where the patch artifacts may occur. We overcome this by averaging the area across the line (as shown in yellow in the last image) in order to make the transition between patches smoother.

In order to prevent degrading the quality of the final generated image, we do not include pixels exclusively from the patch of the second denoising step (pixels that are immediately adjacent to the edge of the random patch from the first denoising step and fall within the second patch) as they primarily contain noise and this degrades the quality of the entire image.

In Figure 8, for illustration purposes, we keep the higher-resolution at $1536^2$. The Slider here is at position 0 as we want to encourage the algorithm to create artifacts that are easily distinguishable, as a stronger guidance results in them being less noticeable. We only use three inference steps here to keep the comparison as close to the base resolution as possible.

## D  Multi-Scale Duplications

Existing patch diffusion approaches risk introducing artifacts with each denoising patch. This phenomena is more prominent at higher image resolutions as the increase in the image resolution leads to an increase in the size of the latent space while the size of the denoising patch still remains constant. Duplications are especially prominent in approaches like Du et al. [2023] where a fixed number of intermediate resolution steps may amplify this issue progressively with every step. We demonstrate this in Figure 10. Although we appreciate that using many steps helps refine the image gradually, depending on the generation the steps may have the negative effect of progressively amplifying a generated artifact at a lower resolution. For this reason we created Pixelsmith to be flexible in its image generation capabilities. Depending on the required result and text-prompt, we allow generating a higher-resolution image without generating any intermediate resolutions first. Image generation at intermediate steps (see also Appendix E is treated as a hyperparameter in our framework.

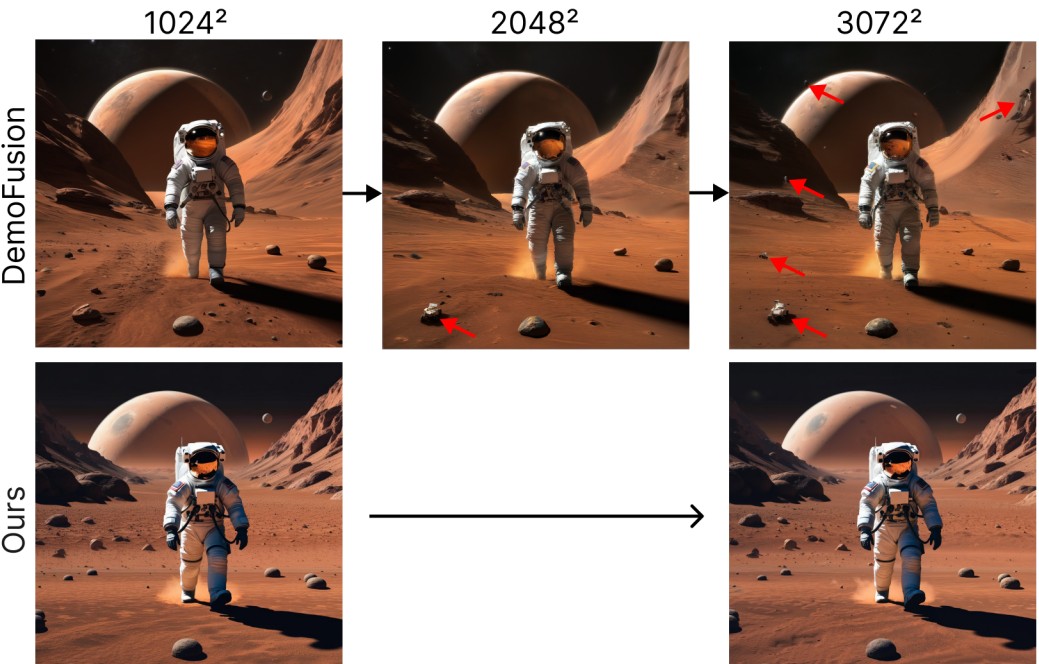

Figure 10: Comparison between a state-of-the-art method, DemoFusion Du et al. [2023] and our work, Pixelsmith. This figure shows how artifacts are amplified at higher-resolutions in DemoFusion (as denoted by the red arrows). We directly generate a $3072^2$ image from the base without any artifacts. The flexibility of our approach allows for any resolution to be generated right after the base resolution minimizing seeing artifacts at every intermediate step (**zoom in** to see in better detail).

# E   Multi-Step Generation

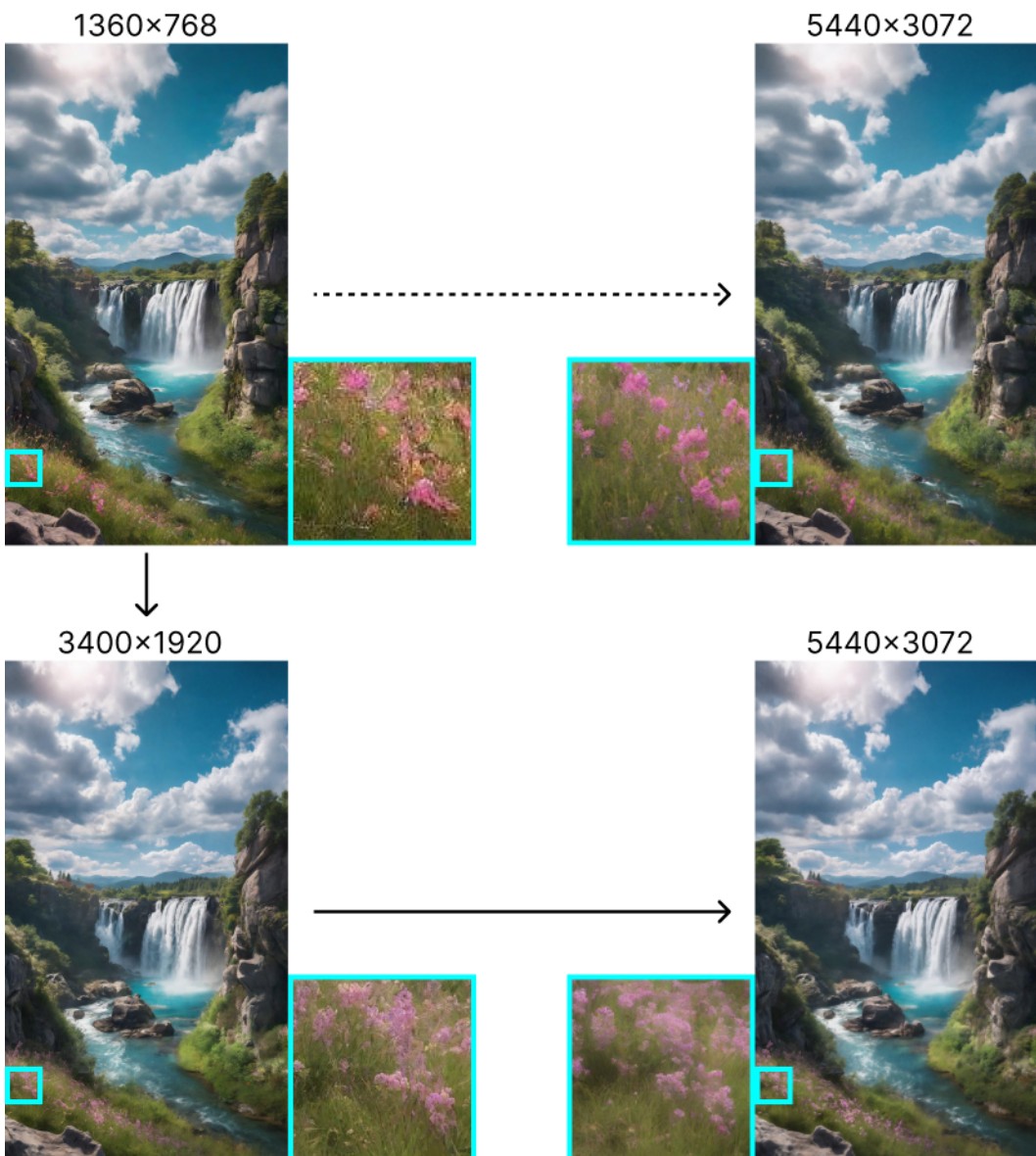

Figure 11: A comparison between two-step and one-step generations. On the top, we show that the base model first generates the image following which we generate the final resolution we desire. At the bottom, we use an intermediate step for better local image feature refinement. From the amplified region of the generation where we employ the intermediate generation step, we see that all the steps contribute to enhancing the features in the image to create a sharper image at the higher-resolution (**zoom in** to see in better detail).

In Pixelsmith, generating the required resolution directly after the base model helps with removing multi-scale duplications of unwanted artifacts (Appendix D). However, allowing to generate multiple intermediate higher-resolution images before the required resolution also leads to a much better definition of objects present in the image at the higher-resolution. We demonstrate this with an example, as shown in Figure 11. We generate a $5440 \times 3072$ image from a base image of resolution $1360 \times 768$ directly, as well as going through the intermediate step of generating a $3400 \times 1920$ resolution image before the final resolution. We amplify the same region in all generated images to show the advantage of having an intermediate generation. Both $5440 \times 3072$ images are highly

detailed, but comparing the amplified region (**zoom in** the image to see in more detail), we can clearly see the high frequency components in the image, such as the grass, are a lot more well defined even though they appear as pixelated regions in the base image. There can however, be cases where this difference in details in the images are not needed/noticeable and a one step image generation would be the preferable and faster choice in such a case.

## F   Super-Resolution Comparison

In addition to comparing our performance with current state-of-the-art models for higher-resolution generation, we also compare our results with a diffusion super-resolution model. Following Scale-Crafter by He et al. [2023], we compare our results with a latent upscaler, but as our base model generates images at a $1024^2$ resolution, choosing the $4\times$ upscaler leads to creating a lot of computational overhead. Hence, we choose the $2\times$ upscaler Rombach et al. [2022a] and evaluate on the $2048^2$ image resolution. Table 3, shows that even though the super-resolution model is trained to upscale, we still outperform it across all metrics.

Table 3: Quantitative comparisons with the $2\times$ upscaler trained for the task of super-resolution. The final image resolution for both approaches is $2048^2$. Pixelsmith outperforms the super-resolution model even in a zero-shot setting.

| Res. | Metric | $2\times$ upscaler | Ours |
|---|---|---|---|
| $2048^2$ | FID↓ | 63.913 | **63.116** |
| | KID↓ | **0.002** | 0.002 |
| | IS↑ | 19.603 | **24.242** |
| | CLIP↑ | 32.250 | **33.429** |

Super-Resolution                    Ours

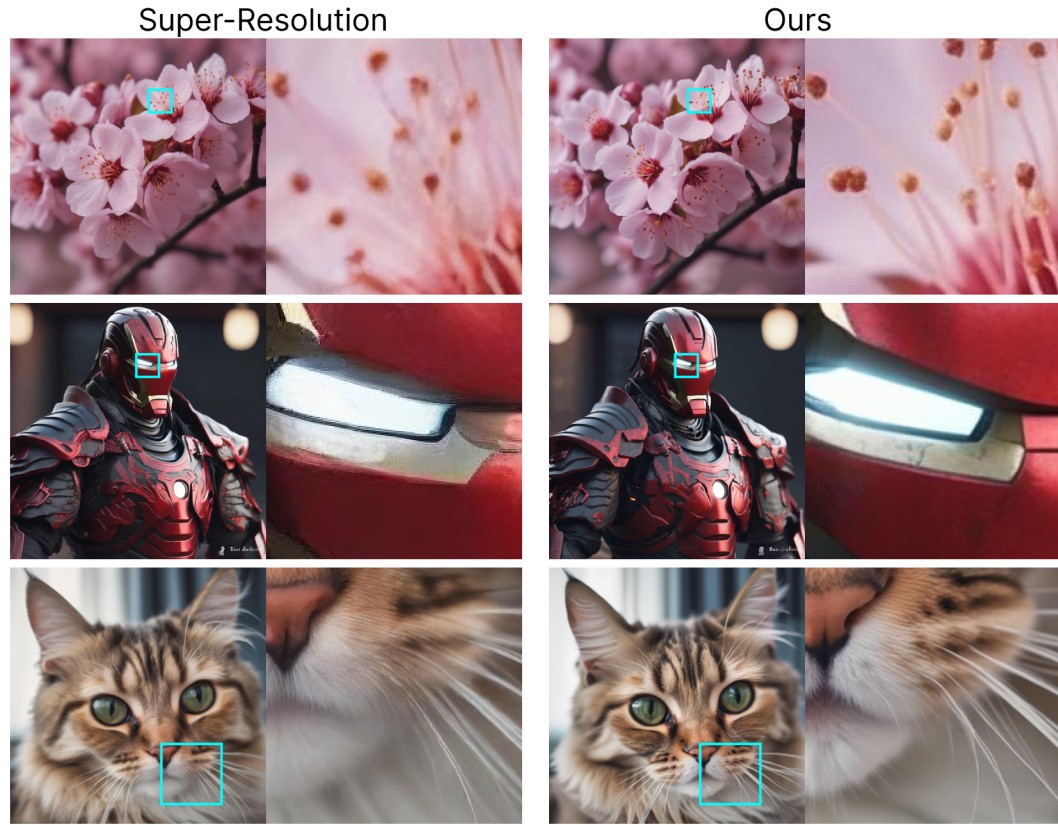

Figure 12: Qualitative comparison with the latent upscaler. The fine details generated by our framework are richer in comparison to the super-resolution model.

# G Slider Position

A generated image depends on many factors, namely, the input text-prompt, the input negative text-prompt, the seed, and the number of inference steps to name a few. The generated images can differ across these factors and even across intermediate image resolutions in terms of shape, texture, colour and a plethora of other varying attributes. This is especially seen when using a model that was not trained on the data for the task at hand directly - making the task a lot more challenging. Varying sizes of objects in the image such as arbitrary big structures, differ from the smaller ones and miscellaneous textured surfaces in the images are different from smooth ones. To account for these changes, and to control and constraint image generation at higher-resolutions, we introduce The Slider. The Slider helps provide a varying amount of control to a user to create image generations depending on how much they want the previously generated scene to influence the next generation.

In a real world scenario, this is especially useful as depending on the type of image being generated and the prominence of artifacts that could potentially be introduced in the image, the user can switch between allowing more guidance from the previous resolution, whereas for truly unconstrained image generation, the Slider provides the added benefit of more freedom. We demonstrate the expressive power of The Slider in Figure 13.

We see more bees generated in the image on the left when we position the Slider value at $1$. This results in certain artifacts and changes the structure of the image. At the other extreme, setting the max value of the Slider ($49$) results in preserving the overall structure of the base image, while not introducing any novel aspects into the generated image. Setting the Slider to $27$ allows the framework to trade-off between image generation and preserving the global structure where it generates more realistic look high frequency components like the hair on the bees' body. Depending on the different settings for the Slider we can constraint the image generation to be more free.

We observe from our experiments that lower values tend to generate images that are rich in fine detail but at the same time, are prone to changing the global structure of the image while creating duplications and artifacts. Higher values tend to keep the overall global structure of the image intact, but do not refine the image to create finer details. The Slider is a tunable hyperameter that provides a means to trade-off between freedom and constraints to create the best possible generation of an image from the text-prompt.

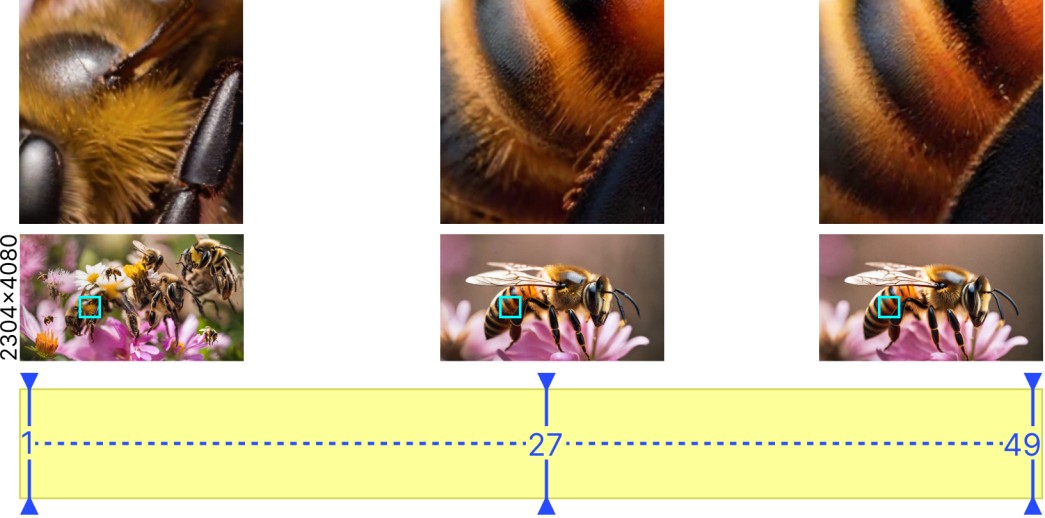

Figure 13: Experimenting with different positions of The Slider. The image on the left contains finer details but loses the overall structure, generating a artifacts. The image on the right has a good global structure but lacks any high frequency details due to the Slider constraining the image generation process. The central position, in this case, is the most optimal generation where the overall structure of the image is preserved and at the same time, finer details are also introduced such as the individual hair strands on the bees' body. We observed that by generating multiple resolutions in a cascaded framework, each intermediate generation will contribute to creating a more detailed final image.

# H   Patch Denoising Masks

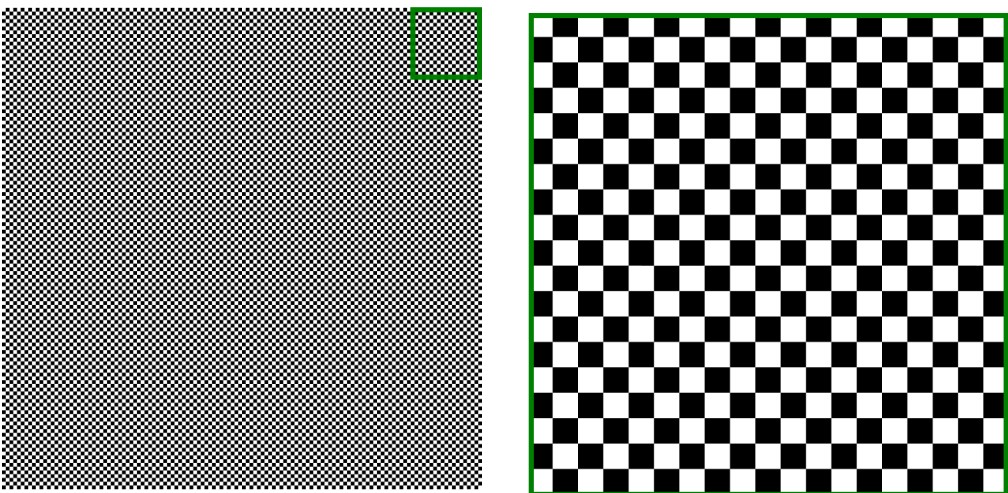

Figure 14: Mask visualisation. Half of the pixels in the denoised patch are masked and replaced with values from the image guidance, significantly reducing duplications.

# I   Additional visualizations

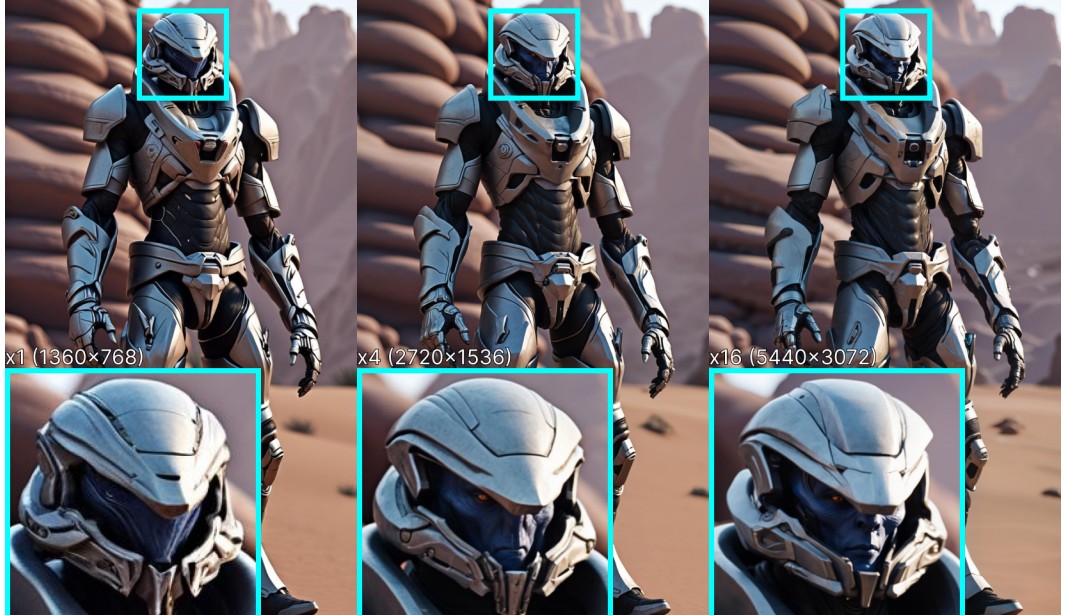

Figure 15: Example of $16\times$ ($5440 \times 3072$) images. The amplified cut-outs illustrate how facial features become increasingly well-defined with each generation (**zoom in** to view in greater detail).

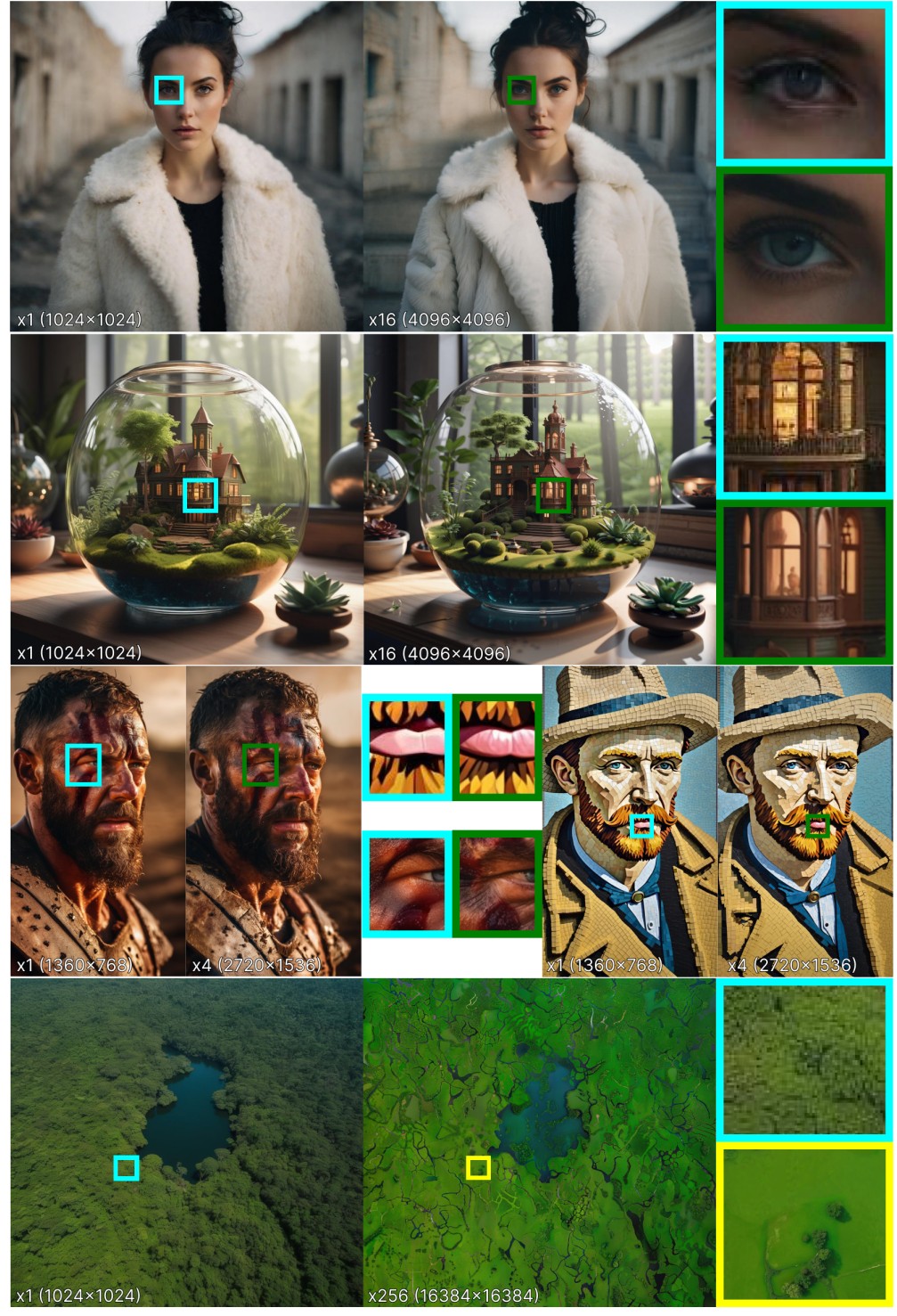

Figure 16: Examples of images at base resolutions alongside their higher-resolution counterparts generated using Pixelsmith (**zoom in** to see in better detail).

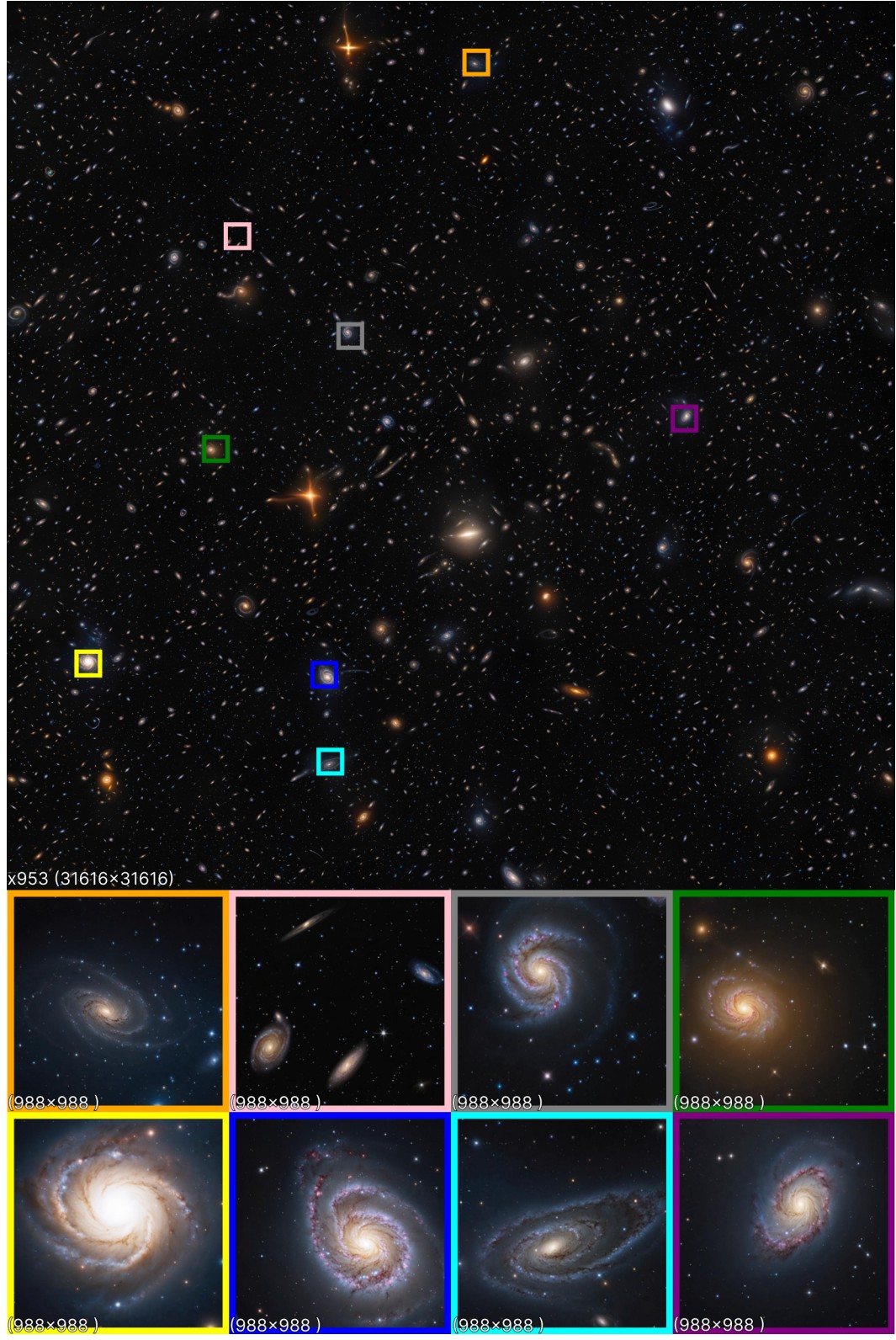

Figure 17: An example of a $953\times$ $(31616 \times 31616)$ gigapixel image. Each displayed patch is $988 \times 988$, roughly the size of the base image. The image closely resembles those captured by the Hubble telescope. (**zoom in** to see in better detail).

## J Text-prompts

Figure 1:

- lego astronaut, riding a lego horse on Mars
- pearly gates of heaven, ethereal, intricate details, angels, cloud, 7th sky, eternal bliss, in an otherworldly universe, detailed, realistic, 8k uhd, high quality
- photo realistic, ultra details, natural light ultra detailed portrait of a female necromancer, skeleton face volumetric fog, Hyperrealism, breathtaking, ultra realistic, ultra detailed, cyber background, cinematic lighting, highly detailed, breathtaking, photography, stunning environment, wide-angle
- Landscape of a dark far away planet, rock and organic soil, glowing trees, UHD, masterpiece, trending on artstation, sharp focus, studio photo, intricate details, highly detailed, by Greg Rutkowski
- photo of the dark sky with visible but very distant swirling galaxies and stars

Figure 2:

- cat, close-up

Figure 4:

- A village on a cliff, realistic transparent sea, fishing boats

Figure 5:

- Photo of a zebra in a forest, black and white
- Photo realistic oil painting of Robert De Niro

Figure 6:

- photo of Emma Watson, on a bike

Figure 7:

- Darth Maul in a white tuxedo, in a forest, photo realistic
- Raw photo of a roman warrior looking at the camera, close-up portrait, high detail, masterpiece
- ultimate Japanese style, ink wash. Black and White, japanese Writing in Red and Blue, trending on ArtStation , intricate details, masterpiece, best quality, Use Dream Diffusion Secret Prompt, Epic

Figure 8:

- Close up photo of the human eye iris

Figure 10:

- Astronaut on Mars

Figure 11:

- masterpiece, best quality, high quality, extremely detailed CG unity 8k wallpaper, scenery, outdoors, sky, cloud, day, no humans, mountain, landscape, water, tree, blue sky, waterfall, cliff, nature, lake, river, cloudy sky, award winning photography, Bokeh, Depth of Field, HDR, bloom, Chromatic Aberration, Photorealistic, extremely detailed, trending on artstation, trending on CGsociety, Intricate, High Detail, dramatic, art by midjourney

Figure 13:

- close-up photo of a bee, macro, 100mm lens, bokeh, depth of field, high quality

Figure 12:

- samurai iron man, dark black samurai armor, Artgerm, photorealism, Unreal engine 5 highly rendered, glowing red eyes, Tilt-shift 4k,White Balance, High Contrast, Low Saturation, Bracketing Detailed.
- macro photography of a sakura blossom flower, epic
- photo of a cute fluffy cat, close-up, high detail

Figure 15:

- a turian on andromeda, 50mm lens

Figure 16

- portrait photo of a beautiful Greek girl, with black pigtails, wearing fluffy white coat, wide mid shot, derelict building, soft lighting, Hasselblad, Voigtländer 50mm lens, in style of Oleg Oprisco
- photography in the style of detailed hyperrealism ,cinematic composition, dramatic light, a glass with a forest in it, surrealism 8k, graphic of enchanted terrarium, 4k highly detailed digital art, 3D render digital art, 3D render stylized, stylized 3D render, highly detailed scene, 3D landscape, 3D landscape, 4 k surrealism, sci-fiish landscape, 3 D artistic render, surreal 3D render. Dreamlike, mysterious, provocative, symbolic, intricate, detailed, expressive,hyper detailed,intricate,poster,artstation
- The gladiator stands after the battle, fear and horror on his face, tired and beaten, sand on his face mixed with sweat, an atmosphere of darkness and horror, hyper realistic photo. In post-production, enhance the details, sharpness, and contrast.
- aerial photo, amazon forest
- mosaic of van Gogh

Figure 17:

- photo of the dark sky with visible but very distant swirling galaxies and stars

