# OpenReview forum: "Is One GPU Enough? Pushing Image Generation at Higher-Resolutions with Foundation Models."
_NeurIPS.cc/2024/Conference — NeurIPS 2024 poster_

### Official Review · Reviewer_snt9 · 2024-07-04

**Soundness:** 3
**Presentation:** 3
**Contribution:** 2
**Rating:** 5
**Confidence:** 4

**Summary:**

In this paper the authors propose a number of engineering tricks which enable generating at higher resolutions from a pre-trained txt2img diffusion model.
Notably, the requirements for the proposed method are relatively low.

In the proposed approach, an image of a standard resolution is generated first.
After that, is it upsampled (in RGB space) to the desired size and used as a guidance for the new, truly high-resolution image which is generated as a sequence of overlapping patches.
The guidance mechanism is implemented by mixing the imaginary component of Fourier decomposition of the currently denoised and guidance patches.
This procedure takes place not for all the denoising steps, but only up to certain noise level referred to as Slider.
The value of Slider controls how similar are the original standard-resolution image and its higher-resolution version.

Performance of the method is evaluated with commonly used metrics such as FID, KID, IS and CLIP-similarity.
Also, a lot of samples are provided for visual inspection.

**Strengths:**

According to the provided metrics and visual results, the methods performs quite well in comparison with recent baselines.
The fact that it requires only 7.4 GB of memory makes it very affordable for the community.
The idea of mixing in the spectral domain, to the best of my knowledge, is novel enough in context of ultra-high resolution sampling from a pre-trained model.
Exploiting the checkerboard pattern for mixing with guidance is also an interesting engineering trick.

**Weaknesses:**

1. While the authors several times mention that "the imaginary part in the frequency space contains most of the low-frequency information of the image" (line 187) and "imaginary part provides more structural information than the real part" (line 249), they do not provide any references for this statement or supporting experiments. To me, this fact does not look evident enough. I am sure this needs more thorough explanation since otherwise it looks just like an empirical trick.

1. From the text, it is unclear which Slider value was used to obtain the numbers from Table 2. Was it the value of 30, as defined in line 244? Also, what exactly is "using next inference step (NIS)" (see line 253)? Is this about checkerboard mask mixing?

1. The presentation of Fig. 3 is not easy for understanding, I advise considering its redesign. For example, forward diffusion and taking imaginary part of FFT are denoted in the same way.

1. It is unclear from the text what the guidelines are for selecting the size for the zone of averaging for overlapping patches (line 805).

**Questions:**

I ask the authors to address the weaknesses listed above during the rebuttal period. In particular, I am interested in the deeper justification of  using imaginary Fourier coefficients than empirical evidence.

POST-REBUTTAL UPDATE: I keep my initial score. I think that although the method is very simple, this does not mean that the submission is bad. I believe that with improved presentation this paper can be interesting for the practitioners.

**Limitations:**

No actions needed.

---

> ### Author Rebuttal · Authors · 2024-08-07
>
> Thank you for the constructive criticism. We welcome any comments that can help improve the quality of our work.
>
> >W1. While the authors several times mention that "the imaginary part in the frequency space contains most of the low-frequency information of the image" (line 187) and "imaginary part provides more structural information than the real part" (line 249), they do not provide any references for this statement or supporting experiments. To me, this fact does not look evident enough. I am sure this needs more thorough explanation since otherwise it looks just like an empirical trick.
>
> Please refer to the global response.
>
> >W2. From the text, it is unclear which Slider value was used to obtain the numbers from Table 2. Was it the value of 30, as defined in line 244?
>
> Yes, the position of the Slider was 30. This is also indicated by the metrics in Table 1 and Table 2 which are identical. Nevertheless, we understand that it should be specifically mentioned in the Comparison section, as such Line 264 is expanded as follows "... refer to Appendix B. The Slider position is set to 30.".
> We note here that the position is not cherry picked but chosen randomly. Different Slider positions will yield different results (as shown in Table 1), and there may be other positions that could potentially perform even better.
>
> >W1. Also, what exactly is "using next inference step (NIS)" (see line 253)? Is this about checkerboard mask mixing?
>
> Yes, this is correct. The "next inference step" mentioned in the Ablation study refers to the use of the chess-like mask. As noted in line 253, "... in Appendix G, incorporating the next inference ...". Appendix G provides further explanation of this process. We recognize that this may have been confusing, so the issue has been addressed as follows:
> * The line 241 is changed to "... information from the next inference step using masking and the impact ..."
> * The paragraph starting with line 252 is changed to " **Masking (column No Mask):** Analysis in column No Mask shows that not using the chess-like mask yields better scores on two metrics. However, as shown in Appendix G, incorporating the mask is crucial for removing artifacts, so we continue to use it despite the metrics."
> Please refer to our reply to reveiwer wkBK for the fully revised Ablation study subsection.
>
> >W3. The presentation of Fig. 3 is not easy for understanding, I advise considering its redesign. For example, forward diffusion and taking imaginary part of FFT are denoted in the same way.
>
> We have revised Figure 3 to enhance clarity. Please refer to the attached PDF for the updated version.
>
> >W4. It is unclear from the text what the guidelines are for selecting the size for the zone of averaging for overlapping patches (line 805).
>
> This is a set number (not a hyperparameter) that we set to 10 and never change. From our experiments, the exact number is not critical, which is why line 805 refers to a "few pixels." We have set this number to 10, and it remains fixed, as variations like 15 or 20 pixels do not produce noticeable differences. This can also be seen in the provided code (smoothed_time_mask = create_gradient_border(time_mask, gradient_width=10)). We understand that specifying the exact value is preferable, so line 804 has been updated to read: "... a tolerance of 10 pixels in the overlap ...".

---

> > ### Comment · Reviewer_snt9 · 2024-08-09
> >
> > I would like to thank the authors for their feedback.

---

### Official Review · Reviewer_T7y4 · 2024-07-10

**Soundness:** 4
**Presentation:** 3
**Contribution:** 3
**Rating:** 6
**Confidence:** 4

**Summary:**

This paper proposes a training-free method for diffusion models to sample images at higher resolution with limited GPU memory, which introduces several tricks, such as fourier merging, chess-mask deduplication, and slider control. Experiments show the effectiveness of the proposed method.

**Strengths:**

1. The fourier merging seems interesting with imaginary part to maintain the global structure;
2. The chess mask seems effectively eliminates the duplicate artifacts in high resulotion generation;
3. The ablations are sufficient and comparisons show the superiority of the proposed method.

**Weaknesses:**

1. Is there any literatures to support that the imaginary part of the fourier transform corresponding to the low-frequency of the signal, or is there any deeper analysis beyond the ablation results in Tab.1 to illustrate this? By the way, it is suggested to provide the visual ablations of this part.

**Questions:**

1. How to determine the patch orders in high-resolution generation, is it random orders or follow some principles.
2. From Tab. 2, it is not clear why Pixelsmith slower than Scalecrafter at 2048 resolution, while nearly 2x faster at 4096 resolution. Is there any trend that the bigger, the more efficient?
3. Is the content of the final scaled high resolution output significantly differ from the base output, whether the new objects will be generated?

**Limitations:**

The authors have adequately discussed the limitations of their work.

---

> ### Author Rebuttal · Authors · 2024-08-07
>
> Thank you for the constructive criticism. We welcome any comments that can help improve the quality of our work.
>
> >W1. Is there any literatures to support that the imaginary part of the fourier transform corresponding to the low-frequency of the signal, or is there any deeper analysis beyond the ablation results in Tab.1 to illustrate this?
>
> Please refer to the global response.
>
> >W1. By the way, it is suggested to provide the visual ablations of this part.
>
> Including visual ablations is indeed a good suggestion. They will be part of the final version.
>
> >Q1. How to determine the patch orders in high-resolution generation, is it random orders or follow some principles.
>
> The patch orders are random, as described in Lines 143 and 183. However, this randomness is controlled. We track which areas have been denoised (since each pixel is denoised only once per timestep) and avoid selecting those areas again. Therefore, while the selection of patches is random, it is constrained to areas that still require denoising. For example, if 90% of the latent space has already been denoised for timestep t, the coordinates for the next patches will be chosen from the remaining 10%.
>
> If we selected coordinates purely at random without considering the denoised areas, the inference time would increase dramatically, rendering the process infeasible, especially for very high resolutions such as $32768^2$.
>
> This explanation will be incorporated into the "Patch Sampling" subsection for added clarity.
>
>
> >Q2. From Tab. 2, it is not clear why Pixelsmith slower than Scalecrafter at 2048 resolution, while nearly 2x faster at 4096 resolution. Is there any trend that the bigger, the more efficient?
>
> We extract patches of the same size, specifically $128²$, regardless of the resolution. Pixelsmith requires 130 seconds for $2048^2$ (4194304 pixels) and 549 seconds for $4096^2$ (16777216 pixels). This represents a 4x increase in pixels and a 4.22x increase in inference time. Pixelsmith’s inference time scales linearly with the number of pixels because the denoising UNet consistently uses patches of the same size. In contrast, Scalecrafter scales differently as it does not use patches. Unfortunatelly, Scalecrafter's official code provides scripts for specific resolutions and no further experiments can be conducted to examine how they scale at different resolutions and most importantly there is also the constraints of the required memory. Pixelsmith apart from not being constrained by memory is very flexible. It is posible to generate an image with resolution 4096x1024 using a $1024^2$ base image. The final ratio is not constrained by the base one. To accomplish this we only change the final resolution (ie just one number), no script or otherwise optimization is needed.
>
> >Q3. Is the content of the final scaled high resolution output significantly differ from the base output, whether the new objects will be generated?
>
> Several factors influence the comparison between the final image and the base image. Key factors include the Slider position, the number of intermediate steps, the resolution of the final image, and the specific text prompt used. The Slider is designed to offer more control over the generation process. For instance, Figure 11 demonstrates how different Slider positions produce varying results, with position 49 being closest to the base image and position 1 showing excessive repetitions. However, in certain contexts, such as generating the surface of a planet with craters, having more craters can be beneficial. Thus, the final content can vary significantly or minimally, and this can be easily adjusted using the Slider.

---

> > ### Comment · Reviewer_T7y4 · 2024-08-14
> > **Official Comment by Reviewer T7y4**
> >
> > Thanks for the authors rebuttal. Most of my questions have been addressed, however, the reason why the imaginary part of the fourier transform corresponds to the low-frequency of the signal is still not illustrated well with deeper analysis, and depend on the ablation experiments, which may undermining the potential inspiration, therefore, I will not raise my score.

---

> > > ### Author Response · Authors · 2024-08-14
> > >
> > > Thank you for the comment! We appreciate the opportunity to continue the discussion.
> > > >Most of my questions have been addressed, however, the reason why the imaginary part of the fourier transform corresponds to the low-frequency of the signal is still not illustrated well with deeper analysis, and depend on the ablation experiments, which may undermining the potential inspiration
> > >
> > > As stated in our global response, all the references have been changed to reflect the concerns raised. For example, in Lines 187 and 248, where it was previously mentioned that the imaginary part of the Fourier transform corresponds to the low-frequency information of the signal, this has now been removed. In our revised paper, we do not state a connection between the imaginary part and the low-frequency information and we base our choice of the imaginary part on the results of the ablation experiments.

---

### Official Review · Reviewer_wkBK · 2024-07-13

**Soundness:** 2
**Presentation:** 1
**Contribution:** 2
**Rating:** 3
**Confidence:** 4

**Summary:**

The paper introduces Pixelsmith, a framework designed to utilize pre-trained diffusion models to enable high-resolution image generation using only a single GPU.
Patch-based denoising ensures that the entire generation process can be accommodated on a single GPU.
The Slider mechanism balances the trade-off between finer details and overall structure by controlling the transition from guided generation to native generation.
Guided generation operate on the latent fused by one upscaled higher-res image patch and one native-generated higher-res image patch in Fourier space, helping to maintain global structures as claimed by the authors.
Experimental results show that Pixelsmith not only produces high-quality and diverse images but also improves sampling time and reduces artifacts compared to existing techniques.

**Strengths:**

1. Memory Efficiency with Limited Computational Resources: Achieving high-resolution image generation on a single GPU is a good contribution, and the image generation quality is not compromised by this restricted setting.
2. High-Quality Generation: The proposed Fourier space fusion maintains fine-grained details and global structures, while also preventing some artifacts that occur in other methods.

**Weaknesses:**

1.	Paper Structure Issues: The content before the methods section is too lengthy, making the key method and experiments sections comparatively short and harder to fully understand. These are elaborated upon in the following points.
2.	Method Description is Hard to Follow: For instance, the overview in Lines #171-173 does not fully describe all the components involved in the method, requiring readers to review the paper multiple times to understand the entire pipeline. Additionally, the intuition behind combining $\hat{z}^{iFFT}{t-1}$ and $\hat{z}^{guid}{t-1}$ is not explained, which is confusing since $\hat{z}^{iFFT}_{t-1}$ already incorporates information from $\hat{z}^{guid}_t$.
3.	Errors and Mismatches in the Experiment Section: According to Table 1, using the real part is better, which contradicts Line #248. Line #252 describes the wrong column for "the next inference step". Also, the term “base model” in Table 2 is unclear—does it refer to the same model as Table 2’s Pixelsmith and Figure 1’s base model?
4.	Flaws in Figure Illustrations: In Figure 1, the base model image for “x256 (16384x16384)” is not displayed. Is this base model the same as the one mentioned in Table 1? In Figure 3, the illustration should be more concrete and easier to understand; however, some variable names are too small and there are too many blank areas. The fusion part is also unclear, with no mention of the real part of the Fourier transform and no explicit explanation of $\mu( . , . )$.

**Questions:**

The aim of achieving high-resolution image generation on a single GPU is commendable, and the proposed method successfully accomplishes this goal. However, the presentation quality is relatively poor, especially with some key parts described incorrectly. I hope the authors can address the questions mentioned in the weaknesses section and resolve my concerns.

**Limitations:**

The authors have discussed the trade-off between achieving finer details and suppressing artifacts. They have also suggested proposing appropriate metrics for evaluating high-resolution image generation, which would be beneficial for the community.

---

> ### Author Rebuttal · Authors · 2024-08-07
>
> Thank you for the constructive criticism. We welcome any comments that can help improve the quality of our work.
>
> >W1. (full question)
>
> We have restructured the content based on the suggestions. The Related Work and Foundations sections have been shortened. The Method section has been revised, as detailed in the global response. Additionally, the Experiments section has been expanded to enhance comprehension as follows:
>
> > ### Ablation study
> > We conduct a qualitative examination of the framework on $2048^2$ image resolution. Specifically, we assess the effects of the Slider position, the importance of the imaginary part, the significance of incorporating information from the next inference step using masking and the impact of averaging overlapping patches.
> >
> > **Slider Position (columns SP0, SP24, SP49):** Our findings indicate that the Slider position significantly influences the results. The proposed model with a Slider position of 30 (Table 1, column Proposed) outperforms positions 0, 24, and 49 (Table 1, columns SP0, SP24, and SP49). A position of 0 introduces numerous artifacts, while a position of 49 lacks fine detail. Position 24 is close to the proposed but not optimal for the random subset. Position 30 was chosen randomly and is not cherry-picked. Appendix F demonstrates the effect of the position with a qualitative example.
> >
> > **Imaginary Part (columns Re, Re\&Im):** The proposed model (Table 1, column Proposed) averages the imaginary parts of the guidance latents and the current latents, then uses the real part of the current latents as described in the Method section (see Figure 3). This setup is chosen based on experimental results demonstrated here. We compare this with averaging the real parts of the two latent spaces and using the imaginary part of the current latents to invert back to the pixel space (Table 1, column Re), as well as averaging both the imaginary and real parts from the two latent spaces (Table 1, column Re&Im).
> >
> > **Masking (column No Mask):** Analysis in column No Mask shows that not using the chess-like mask yields better scores on two metrics. However, as shown in Appendix G, incorporating the mask is crucial for removing artifacts, so we continue to use it despite the metrics.
> >
> > **Averaging (column No Aver.):** Finally, we show that using patch averaging (Table 1, column Proposed) improves scores compared to not using it (Table 1, column No Aver.). This is because patch averaging eliminates patch artifacts, as seen in Appendix C.
> >
> > Table1: A quantitative examination of our framework through ablations.
> >
> >| Metric        | SP0    | SP24   | SP49   | Re     | Re&Im  | No Mask | No Aver. | Proposed |
> >|---------------|--------|--------|--------|--------|--------|--------|----------|----------|
> >(unchanged)
>
>
> >W2. (full question)
>
> We have revised the Method description to better align with Figure 3 (please see the global response). Additionally, we have updated Figure 3 (please see the attached PDF) to enhance readability and coherence. Regarding the second part of your concern, please refer to Appendix G for a qualitative comparison and the Experiments section for a quantitative comparison. Every time a denoising step finishes, there is a chance for artifacts to appear. The chess-like mask is applied after the denoising, replacing some information with the guidance where no artifacts exist.
>
>
> >W3. Errors and Mismatches in the Experiment Section: According to Table 1, using the real part is better, which contradicts Line \#248.
>
> In Lines \#247-248 we state that "Using the imaginary part to average the guidance latents and the current latents is more effective than using the real part or...". This statement is consistent with the results shown in Table 1, where all four metrics demonstrate better performance when using the imaginary part (last column) compared to using the real part (column Re).
>
>
> >W3. Line \#252 describes the wrong column for "the next inference step".
>
> You are correct, this was an error. Thank you for bringing this to our attention. We have now corrected it.
>
> >W3. Also, the term “base model” in Table 2 is unclear—does it refer to the same model as Table 2’s Pixelsmith and Figure 1’s base model?
>
> Throughout the paper we refer to SDXL as the base model (one of the references is in Line \#258: "SDXL, which serves as the base model"). In the Experiments section, we also referred to our final proposed model as the base model, which caused confusion. This inconsistency has now been corrected by replacing the base model in the Experiments with proposed.
>
> >W4. Flaws in Figure Illustrations: In Figure 1, the base model image for “x256 (16384x16384)” is not displayed.
>
> The higher resolution images are in scale with those generated by the base model. In the “x256 (16384x16384)” case, the base model image is not missing but hard to see due to the significant difference in resolution. As with the other images in Figure 1, the base model image is located at the bottom-right, overlaid on the higher resolution image. We understand this may confuse readers, so we have added a white box around all base model images to make them more distinguishable. The legend of Figure 1 has been expanded as follows: "... Pixelsmith and the base model. The higher resolution images are in scale with the images generated by the base model. The base model generations are enclosed in a white frame. Some cut-outs ..."
>
> >W4. Is this base model the same as the one mentioned in Table 1?
>
> Please refer to our response to W3.
>
> >W4. In Figure 3, the illustration should be more concrete and easier to understand; however, some variable names are too small and there are too many blank areas. The fusion part is also unclear, with no mention of the real part of the Fourier transform and no explicit explanation of.
>
> We have revised Figure 3 (please see the attached PDF). For a detailed explanation, please refer to the global response.

---

### Official Review · Reviewer_SvAB · 2024-07-18

**Soundness:** 2
**Presentation:** 1
**Contribution:** 2
**Rating:** 4
**Confidence:** 5

**Summary:**

This paper introduces a framework for generating high-resolution images from text prompts using pre-trained diffusion models. The key innovations are: A cascading approach that uses lower-resolution generated images as guidance for higher resolutions. A "Slider" mechanism to control the balance between following the guidance and allowing novel generation. Patch-based denoising to enable generation of arbitrarily large images on a single GPU. Averaging techniques to reduce artifacts from patch-based generation. The authors demonstrate that Pixelsmith can generate images up to 32,768 x 32,768 pixels on a single GPU, outperforming existing methods in terms of quality and efficiency.

**Strengths:**

The proposed method enables generation of ultra-high resolution images without additional training, addressing an important limitation of current models.

Good results.

The patch-based approach allows for generating massive images on consumer GPUs, which is a significant practical advantage.

**Weaknesses:**

1. Poor presentation. The writing of this paper needs substantial improvement to be published in a venue like NeurIPS. First, the language of the paper needs to be improved. Second, the paper spends a lot of space on unnecessary content. The method in this paper is not difficult, but it is very difficult to understand the method in one reading. In terms of writing, the authors did not succeed in emphasizing the core concept of their proposed method. In addition, the length of the paper is too long. Such a simple method, but the experiments are not introduced until page 8, which should not happen.
2. The core idea of ​​this paper seems to be to generate once and then upsample the generated image by patch-wise processing. (By the way, Figure 3 is difficult to understand and Eq. 4 is not explained). This method is essentially not much different from other operations that use diffusion for upsampling. So what is the advantage of this method? Why is this method feasible?
3. The role of FFT Transformation is not well demonstrated. No exploratory experiments are used to demonstrate the motivation and effect of this method.

**Questions:**

See Weaknesses please.

**Limitations:**

The authors claim they discussed the limitation but I didn't find it in the text. Correct me if I am wrong.

---

> ### Author Rebuttal · Authors · 2024-08-07
>
> Thank you for the constructive criticism. We welcome any comments that can help improve the quality of our work.
>
> >W1. Poor presentation. The writing of this paper needs substantial improvement to be published in a venue like NeurIPS. First, the language of the paper needs to be improved. Second, the paper spends a lot of space on unnecessary content. The method in this paper is not difficult, but it is very difficult to understand the method in one reading. In terms of writing, the authors did not succeed in emphasizing the core concept of their proposed method. In addition, the length of the paper is too long. Such a simple method, but the experiments are not introduced until page 8, which should not happen.
>
> We carefully review and change the language to make it easier for readers to follow. The method section has been refined to highlight the core concept effectively. Additionally, we have edited the method figure to enhance the clarity. See 'global rebuttal' and the pdf with the Figure. Could you kindly provide specific examples where the language was not up to the NeurIPS level as it will help us better revise our paper?
> The "Related work" and "Foundations" are shortened so that the "Experiments" can be introduced at page 7. Given the nature of the paper images take a lot of space (for example Figure 1 takes a full page) and as a result sections are pushed back.
>
> >W2. The core idea of this paper seems to be to generate once and then upsample the generated image by patch-wise processing. (By the way, Figure 3 is difficult to understand and Eq. 4 is not explained). This method is essentially not much different from other operations that use diffusion for upsampling. So what is the advantage of this method? Why is this method feasible?
>
> We have revised Figure 3 to make it easier for readers to understand. As mentioned in the global response, the revised "Method" section and updated Figure 3 are now better aligned, ensuring improved coherence.
> Our method is fundamentally different from diffusion models used for upsampling. While diffusion models trained specifically for super-resolution generate high-resolution images from low-resolution inputs, our approach generates high-resolution images directly from text prompts. Diffusion models for upsampling primarily focus on the input image and largely ignore the text. In contrast, our method relies heavily on the text prompt, which drives the changes. As the patches denoise different areas of the latent space, the content of the text prompt becomes apparent.
> Without our proposed method, numerous duplications would appear across the final image. Our framework restricts these duplications, and depending on the position of the Slider, some or a lot of new information will appear in the higher resolution that is not present in the lower resolution. For example, in Figure 1 (top right), the low-resolution cut-out box for the necromancer woman resembles a metallic connection without a distinct shape. Our method's patch denoising, considering the text prompt, reveals a skull, which makes sense for a necromancer. The patches attempt to denoise based on the text prompt but are constrained by the proposed key components to avoid, for example, denoising multiple necromancers all over the image. Current methods attempting similar results often end up with artifacts, which is not the case with Pixelsmith, as the Slider’s position can eliminate them.
> In conclusion, diffusion models for upsampling lack this generative freedom. The downside of current works is the introduction of duplications and strange artifacts, but Pixelsmith provides control (via the Slider’s position) to avoid them.
>
> This explanation will be reflected in the final version.
>
> >W3. The role of FFT Transformation is not well demonstrated. No exploratory experiments are used to demonstrate the motivation and effect of this method.
>
> The FFT transformation is used to fuse information between the guidance latent space and the current (higher resolution) latent space. The decision to use the imaginary part is based on empirical results. This is now reflected in our paper as mentioned in the global response. The effect is demonstrated in Table 1.
>
> In Table 1:
> - The column labeled **"Re"** represents the case where we averaged the real parts of both latents and used the imaginary part of the current latent.
> - The column labeled **"Re&Im"** indicates that we averaged both the real and imaginary parts of the two latents.
> - The final column shows the proposed method, where we averaged the imaginary parts of the two latents and used the real part of the current latent.
>
> As shown, the proposed method in the last column performs better.
>
>
>
> >L. The authors claim they discussed the limitation but I didn't find it in the text. Correct me if I am wrong.
>
> Please refer to "6 Discussion and Considerations." One of the main limitations in generating higher-resolution images is the availability of effective metrics. More research is needed to develop methods that can accurately evaluate images without reducing their resolution to much smaller sizes, which results in the loss of detail. Additionally, there is no penalty for artifacts and duplications commonly found in many recent papers (see Figure 4 and Figure 6).
>
> In the same section, we discuss Pixelsmith and the trade-off between preserving fine detail and suppressing artifacts. As noted, achieving true higher-resolution detail becomes increasingly challenging as the resolution increases, making it difficult to eliminate artifacts completely.

---

> ### Comment · Reviewer_SvAB · 2024-08-13
> **Response to the rebuttal**
>
> I have read the author's response, as well as the comments and discussions with other reviewers. The author has partially addressed my concerns. However, I still think that the presentation of the paper is lacking at this stage. I will improve my score. However, since I cannot see the revised paper, I cannot judge whether the final presentation meets the requirements of NeurIPS.

---

> > ### Author Response · Authors · 2024-08-14
> >
> > Thank you for the comment! We appreciate the opportunity to continue the discussion.
> > > The author has partially addressed my concerns.
> >
> > Can you please tell us which concerns have not been addressed so we can attend to them as well?
> > > However, I still think that the presentation of the paper is lacking at this stage.
> >
> > In our original reply, we kindly asked for specific examples where the language was not up to the NeurIPS level, as it is important to us to have this constructive feedback. Could we please extend the original request here and ask for specifics on where the paper is lacking as well? We have revised the entire paper and followed all suggestions from all reviewers. It would greatly benefit our work to know which parts are lacking, as we understand that this is the main (remaining) issue with the review. Examples of language before the revision (referring to W1) and the presentation after all the implemented changes would help our work, as we believe in the NeurIPS review process and the best practices to help promote research.
> > > However, since I cannot see the revised paper, I cannot judge whether the final presentation meets the requirements of NeurIPS.
> >
> > As we cannot upload the full paper, I can kindly point you to the Method section, which is in the global response, and the Experiments section, which is a reply to reviewer wkBK. The Related Work and Foundations sections that were changed can be found below. The rest of the paper remains unchanged, so the full paper is now shown in our replies.
> > >
> > > # Related Work
> > > Pre-trained DMs are following a ... each new version [30]. It is clear that there is demand for increasingly higher resolution generation.
> > > Currently, generating images ... application.
> > > ## Trained models
> > > (unchanged)
> > > ## Adapted models
> > > (unchanged)
> > > # Foundations
> > > ## Diffusion models
> > > DMs [15,42] are probabilistic generative models that first add noise to a distribution during diffusion and then learn to remove this noise during denoising. During training, a Gaussian probability distribution is learned, and during inference, sampling from the Gaussian leads to the data probability distribution. Executing this process in the latent space [37] is more resource-efficient, allowing for faster training and inference times.
> > > In formal terms, for a Latent Diffusion Model, if $z_0$ represents the data point in the latent space, given $z_0 \sim q(z_0)$ and $q$ being the diffusion process, then for timesteps t $\in$ $\{1,T\}$ $z_1,\ldots,z_T$ noisy latents, with variance $\beta_t \in (0,1)$, are produced, defining a joint distribution conditioned on $z_0$:
> > >
> > >(unchanged equation 1)
> > >
> > >(unchanged equation 2)
> > >
> > > The training estimates an isotropic Gaussian distribution for $z_T$. During the denoising process we sample from $z_T \sim \mathcal{N}(0,{I})$ and remove the added noise using a neural network:
> > >
> > >(unchanged equation 3)
> > >  ## Patch sampling
> > > The default denoising process of an LDM involves sampling the entire latent space at each timestep. While this approach works for lower resolutions, it becomes increasingly resource-intensive as the resolution increases.
> > > Instead, we modify the default process to denoise patches of a fixed dimension $128^{2}$ as introduced by DiffInfinite [1]. At each timestep, random patches are selected for denoising, and this process is repeated until the entire latent space is denoised (see Appendix Patch sampling). The DiffInfinite process relies on segmentation masks to condition each individual patch, providing rich spatial information. In a text-to-image DM, where the text-prompt is global for the entire latent space, using this method means that each patch is denoised with the same condition—the text-prompt. This leads to multiple repetitions of the condition and results in poor-quality generations. To address this, we implement a series of key components that enable scaling a pre-trained DM to resolutions never achieved before.
> >
> > The above changes result in considerably smaller pre-Method sections, allowing the new, better-explained Method section to fit and enhance readability. The previous full text of this section will become a new appendix, titled "Patch sampling," with the addition of our explanation to question 2 of Rv T7y4. Nothing else changes or is added, so this appendix will not be new to you.

---

### Author Rebuttal · Authors · 2024-08-07

We thank the reviewers (Rvs) for the valuable feedback and the opportunity to improve our work.

## Acknowledged Strengths

**Results:** All 4 Rvs agree that the paper achieves good results compared to current works. It's worth emphasizing that these works are published at the most prestigious conferences, underscoring the significant impact and competitiveness of our results

**Contribution:** Rv SvAB highlights the importance of our work, noting that it addresses an "important limitation of current models". This recognition emphasizes our paper's relevance in advancing the field and tackling key challenges. Further, Rvs SvAB, wkBK, and snt9 acknowledge the importance of our ability to use just 1 GPU regardless of the final resolution, identifying this as a significant contribution. We would like to add that our framework not only is memory efficient but the only one to scale a pre-trained diffusion model up to $32768^2$ ($\approx$1.1 gigapixel). Most current works show results up to $4096^2$ and a very few of them scale up to $8192^2$ which means that our model is able to generate $\times16$ more pixels than any current work

**Originality:**  Rv snt9 commends our method as both novel and interesting

**Ablations:**  Rv t7y4 finds that no further ablations are needed and that the introduced parts in the method are effective and interesting. We note that the rest of the Rvs also do not request extra experiments showing the thoroughness of our evaluation

## Key Concerns

**Fourier transformations:** Some Rvs have noted issues with lines 187 and 248 concerning the justification of using the imaginary part of the guidance latents and why we did not use the real instead. We observed that removing some of the data when fusing the two resolutions of the image helped to improve results and seemed to preserve low-frequency information well. The ablation studies compared Re+Im, Im only, and Re only. Im only performed best empirically. We thank the Rvs for highlighting this, as our observations were poorly formulated. We will change all references to reflect that it is based on empirical observations

**Presentation:**  Some Rvs were concerned about the presentation. We have revised the organization to enhance clarity. The Related Work and Foundations have been reduced and the Method has been rewritten (see below a preview due to character limitations). Additionally, Fig. 3 has been revised to align more closely with the Method, ensuring a seamless understanding (f.i. in Fig. 3 the "3. Image guidance" aligns with "Image guidance" in Overview)

>## Framework
>In this section, we describe the workflow of Pixelsmith, detailing how the framework adapts pre-trained text-to-image LDMs to generate images with higher resolutions on a single GPU (see Fig. 3). In order to generate ultra high resolution images without artifacts, we introduce these key components: the Slider (see Slider), patch averaging (see Patch averaging) and masking (see Masking).
>
>### Overview
>
>**Text-to-image generation:** First, given a conditional prompt $c$, ... (line 174-176) SDXL.
>
>**Upsampling process:** After the image generation, we apply an upsampling algorithm ... (lines 176-179)
>
>**Image guidance:** Once the guidance image is encoded in the VAE's latent space $z_0^{guid}=\mathcal{E_\theta}(x^{guid})$, we can easily sample each latent variable of the diffusion process through the forward diffusion process $z^{guid}_t \sim q(z^{guid}_t|z^{guid}_0)$.
>
>**Image generation:** The generative process starts from $z_T \sim \mathcal{N}(0,I)$, which has the same dimensions as $z^{guid}_T$. At each step, a random patch is cropped as described in Section 3.2:
>
>(Eq. 4)
>
>where $\mathcal{C}^{i,j}$ crops the latent variables $z_t,z_t^{guid}$ at the coordinates $i,j$ for the patch sampling described in Section 3.2.
>
>The Slider’s position (see Slider), indicated by a blue line in Fig. 3, determines whether the guidance mechanism (see Guidance mechanism) or unguided patch denoising will be applied. In the unguided mode, each patch is based solely on the previous one, similar to a conventional patch denoising process. The Slider allows control over whether a generated image will be slightly or significantly altered compared to the previous resolution.
>After the denoising process has ended, the latents $z_0$ are decoded and the higher resolution image is generated. Using a cascade upsampling approach, this generated image can be upsampled again, repeating the process to achieve an even higher resolution image.
>
>### Cascade upsampling
>(unchanged)
>
>## Guidance mechanism
>The guidance mechanism fuses the $\hat{z}^{guid}_t$, $\hat{z}_t$ and $\hat{z}^{guid}\_{t-1}$ random patches to generate the $\hat{z}\_{t-1}$ patch. First, the $\hat{z}^{guid}_t$ and $\hat{z}_t$ patches are transformed to the Fourier space using a Fast Fourier Transformation ($\mathcal{FFT}$), where their imaginary parts $\mathcal{I}m$ are averaged:
>
>(Eq. 5)
>
>The imaginary part is then combined with the real part of the $\hat{z}_t$ patch to form $\hat{z}^{FFT}_t = \mathcal{R}e(\hat{z}_t) + i \hat{z}^{im}_t$
 , which is then transformed back to the spatial domain:
>
>(Eq. 6)
>
>The output is then used as the condition for the reverse diffusion process
>
>(Eq. 7)
>
>To prevent further prompt duplications across the entire latent, we use masking.
>
>### Masking
>
>We combine the sampled $\hat{z}^{iFFT}\_{t-1}$ with the image guidance $\hat{z}^{guid}\_{t-1}$ using a chess-like mask $\Lambda$
>
>(Eq. 8)
>
>### Patch averaging
>
>Due to overlapping patches, visible distinctions can sometimes be noticed at the borders of the patches (Appendix C). To eliminate these, we create a zone where the patches meet and take the average of both patches to achieve a seamless denoised result.
>
>### Slider
>(unchanged)

Masking, Patch averaging and Slider will also include Appendices G, C and F

---

### Decision · Program_Chairs · 2024-09-25

**Decision:**

Accept (poster)

**Comment:**

This paper presents a practical approach to generating ultra-high-resolution images from pretrained text-to-image foundation models. The reviewers find that it addresses a significant limitation of previous work, and the presented paper is the first one to achieve gigapixel image generation on a single GPU, which is a substantial improvement over prior work. The method combines a few different technical designs (e.g., FFT features) for better generation quality. On the other hand, the presentation of the paper needs improvement, and the AC hopes the authors can further revise the paper to incorporate the suggestions of the reviewers. The AC believes that the paper has technical merits with strong experimental results, and the major concerns are addressed. Thus, the AC recommends the acceptance of the paper. The decision of this paper has been discussed with the SAC.